



# Data driven clustering of rain events: microphysics information derived from macro scale observations

Mohamed Djallel Dilmi[1], Cécile Mallet[1], Laurent Barthes[1], Aymeric Chazottes[1]

[1]LATMOS-CNRS/ UVSQ/ UPSay, 11 boulevard d'Alembert, 78280 Guyancourt, France

Corresponding author : Laurent Barthes (laurent.barthes@latmos.ipsl.fr)

Keywords : rain events, rain microphysics, classification, Genetic algorithm, clustering, Self-Organising Maps, unsupervised
learning, Hierarchical clustering

**Abstract**. The study of rain time series records is mainly carried out using rainfall rate or rain accumulation parameters estimated on a fixed duration (typically 1 min, 1 hour or 1 day). In this paper we used the concept of rain event. Among the numerous existing variables dedicated to the characterisation of rain events, the first part of this paper aims to obtain a
parsimonious characterisation of these events using a minimal set of variables. In this context an algorithm based on Genetic Algorithm (GA) and Self Organising Maps (SOM) is proposed. The use of SOM is justify by the fact that it allows to maps a high dimensional data space to a two dimensional space while preserving as much as possible the initial space topology in an unsupervised way. The obtained 2D maps allow to provide the dependencies between variables and consequently to remove redundant variables leading to a minimal subset of variables. The ability of the obtained 2D map to deduce all events
characteristics from only five features (the event duration, the rain rate peak, the rain event depth, the event rain rate standard deviation and the absolute rain rate variation of order 0.5) is verified. From this minimal subset of variables hierarchical cluster analysis were conducted. We show that a clustering in two classes allows finding the classic convective and stratiform classes while a classification in five classes allows refining this convective / stratiform classification. Finally, the last objective of this paper is to study the possible relationship between these five classes and their associated rain event microphysics. Some
relationship between these classes and microphysics parameters are highlighted.

## 1. Introduction

One method to obtain information regarding the characteristics of precipitations at a particular location and for a specific
application is the use of the concept called "precipitation event" or "rain event". Such a concept is a convenient way to summarize precipitation time series in a small number of characteristics so that they make sense for particular applications.

The concept of a precipitation event is not new and has been used for many years (Eagleson, 1970; Brown et al., 1984). A wide variety of definition depending on the context of the study has been investigated in the literature (Larsen and Teves, 2015). Moreover, when a rain rate time series (generally based on a pluviometer record) has been resolved into individual
rainfall events, a wide variety of characteristics of these events such as average rainfall rate, rain event duration and event rainfall distribution (known as hydrological information) can be calculated for each event. We identified seventeen features in the literature to characterize a rain event, this make the comparison between different studies quite difficult. The first goal of this study is to select the most relevant features to characterize the events through a data-driven approach without taking into account the application context and thus to characterize a rain event in the most parsimonious and efficient possible way.



The second objective is to assess, without any a priori, that rain events are still properly clustered from these most relevant observed features. Indeed, the specialists of atmospheric processes distinguish stratiform events from convective events, arguing that physical processes involved in their evolution are different. The goal here is to check that a small sample of variables, derived from point measurements to describe rain events, can realize this distinction and eventually refining it. The hydrological (also called macro physical hereafter) information aims at characterizing rain events from rain gauges measurements. Per se the hydrological information is defined to characterize global event features and is not defined to provide any information about the raindrop microphysics of the event. Nevertheless in many applications such as remote sensing the knowledge of the microphysics is essential. Usually the microphysics is characterised by the raindrop size distribution, noted $N(D)$, which is defined by the number of raindrops per unit of volume and per unit of raindrop diameter ($D$). Actually, information on rain microphysics is often displayed through proxies of $N(D)$ as it will be explained further. At the present time, the rain microphysics features are not accessible by rain gauges which only provide macrophysical information. Only, much more expensive devices called disdrometers provide both hydrological and microphysical information. Around the world, there are few disdrometers where there are tens of thousands of rain gauges. As it will be shown further some of the microphysical information embedded in the hydrological information can be retrieved. In consequence, observations given by rain gauges could be very useful for microphysics studies by providing indirectly through a statistical approach the missing microphysics information. In the following the terms "macro physical" or hydrological information will be associated to characteristics related to rain rates or rain accumulation while the term "microphysical" will be associated to characteristics of the raindrop size distribution.

In this work, we use a data-driven approach to study the relations between different rain properties. Disdrometers provide drop size distribution and consequently they allow estimating one-minute rain rates which are the measurements used to get hydrological information. This information is coherent with what could be provided by standard pluviometers. Having both microphysical and hydrological information we are also able to analyse the microphysical properties of the rain-event clusters provided by our algorithm. This allows considering the possibility of retrieving (unobservable) microphysical information from rain gauge measurements.

From a single rain-rate times series observed with a one minute time resolution, we wish to answer to the following questions : Among the great number of hydrological information variables found in the literature which are the most significant? Does the resulting description of rain events allow discriminating between different types of rain events? What (unobserved) microphysical properties of an event or a type of rain event can be inferred from its macro physical description?

The paper is structured as follows. Section 2 presents the data employed in the study and lists some hydrological information commonly found in the literature. As shown in this section, these 17 identified macro physical variables need to be properly normalised. Section 3 presents the methodology based on a genetic algorithm (GA) relying on the use of a self-organizing map (SOM also called topological map). This unsupervised approach is used to select some of the 17 identified variables. We are hence able to obtain a parsimonious characterisation of the rainfall events. An exploratory statistical analysis of rainfall events is provided. In section 4 the rainfall events are grouped in cluster. At first the rainfall events are divided in two classes and we show that this partition of the data set corresponds to the standard convective / stratiform classification. In a second time a classification in five subclasses is proposed. This classification is a refinement of the previous one. In section 5 we add some microphysical features of the rainfall events allowing studying the microphysical properties of the five event classes previously defined. Finally we conclude the study in section 6.

## 2. Used data sets - Retained approach and Data processing

This work leans on a raindrop collection obtained with a disdrometer and more specifically a Dual-Beam Spectropluviometer (DBS) described in Delahaye et al. (2006). This instrument allows the recording of incoming drops through their arrival time



as well as their diameter and fall velocity. The capture area of this sensor is 100 cm$^2$ so the obtained observations can be considered as punctual. In this study the integration time $T_{int}$ was set to one minute. The raindrop collection is used to estimate the corresponding one-minute rain rates time series $RR_t(t)$. In order to eliminate false raindrop detections that could be generated by dust or insects a threshold $T_0 = 0.1$ mm.h$^{-1}$ is applied. Hence rain rates lower than $T_0$ are set to zero. This conventional threshold is also chosen to maintain the coherence with previous studies (Verrier et al., 2013; Llasat et al., 2001). For this study we worked with two data sets recorded over a period ranging from July 2008 to July 2014 at the "Site Instrumental de Recherche par Télédétection Atmosphérique" (SIRTA[1]) in Palaiseau, France.

### 2.1 Rain event definition

Everyone can observe that rain starts to stop some time later. This is how a rain event is defined in everyday life. However, given the discreet nature of rain (made out of drops) it is not an easy definition. Indeed a rain event will depend on the sensor characteristics (specific surface caption, detection threshold, instrumental noise). This definition may also depend on the purpose of the study and thus on the scientific community behind it. Hence a wide variety of criteria exists to divide precipitation records into rain events. To keep the results as unambiguous as possible the definition of a "rain event" has to be clearly stated.

In this study the pattern given by the one-minute rain rate time series $RR_t(t)$ can be simplified by grouping non-null rain rates into a set of separate "primitive events" (Brown et al., 1985). On the basis of an assigned Minimum Inter-event Time (MIT) (Coutinho et al., 2014), each rain rate corresponding to a particular minute is assigned to a given rainfall event, either the one in progress or a subsequent supposed independent new rainfall event. In fact, the MIT is defined as the duration of a dry period $D_{dry}$ beyond which the occurrence of non-null rainfall marks the beginning of a new event. For dry periods lower than the MIT rain rates from either side of this period are considered to belong to the same "composite event". Authors have proposed different values of MIT that ensure event independence. Llasat (2001) noted that: "*The definition of an episode is quite subjective. In this case it was felt possible to distinguish between two different episodes when the time which elapses between them without rainfall exceeds 1h, which ensures that, the two episodes come from different 'clouds'*". Bocquillon and Moussa (2014) wrote: "*the constant rain observations on less than thirty minutes represent only 5% of all the rainy periods. The representative threshold of the discretization of the data is 30 minutes to an hour.*"

Dunkerley (2008 a, b) carried out an analysis of the Inter-Event Time (IET) in order to check the influence of this variable on the definition of rainfall events and its influence on the average rainfall rate. As highlighted in this study, when determining a value for the MIT the compromise between independence of rain events and intra-event variability of rain rates is crucial. The selection of the MIT directly impacts the estimated macro physical characteristics. Other researchers proposed to use MIT values of 20 minutes, 1hour or 1day (see Dunkerley, 2008a for a detailed list). We decided in this study to set it to 30 minutes. This value is in accordance with what was used by Coutinho et al. (2014), Haile et al. (2011), Dunkerley (2008a, b), Balme et al. (2006) or Cosgrove and Garstang (1995).

Finally with our data set this choice leads to 545 rain events divided in two subsets, i.e. one for the learning and the other for the test (Tab. 1). The training data set is composed of observations collected during two years in 2013 and 2014 with an availability of 96.4% while the test set collected during the period 2008 – 2012 contains periods with missing data due to a dysfunction of the device.

**Table 1: Observation periods and availability of DBS observations and number of rain events for learning and test data sets**

---

[1] http://sirta-dev.ipsl.jussieu.fr/joomla/index.php/85-article-sans-categorie/71-sirta-home-page



### 2.2 Macro physical description of rain events

Rain events contain numerous information that it is desirable to condense to a restricted set of well-chosen features. There is not a suitable or conventional list of the macro physical features needed to describe and summarize all the information relative to an event. Therefore we decided to consider a significant number of features representative of what is found in the literature dealing with macro physical information. The 17 characteristics we identified (Llasat, 2001; Moussa, 1991) are presented in Tab. 2. Some of them are parameter dependent like $P_c$ which uses 3 values of parameter c. These 3 values lead to 3 $Pc$ indices namely $Pc_1$, $Pc_2$, $Pc_3$. Finally a total of 23 indicators were defined and numbered from 1 to 23 (column 1 in Tab. 2).

**Table 2: The 23 variables identified in the literature to characterize a rain event**

Among the 23 indicators (also called variables hereafter) corresponding to the previous defined features some are very classical like the event duration ($D_e$), the quartile ($Q_i$), the mean event rain rate ($R_m$) or the standard rain rate deviation ($\sigma_R$). Some others are less traditional like the $\beta_L$ parameter (indicator of the convective nature of the rain see Llasat (2001)), the absolute rain rate variation of order c ($P_c$) or the absolute rain rate variation ($P_{s,c}$). Some variables usually used to describe times series like fractal dimension, trend, seasonality, autocorrelation, multifractal parameters require long series of data and are not well suited for an event by event analysis. Even if, for events composed of very few samples (very low value of variable $D_e$) computation of some indicators ($\sigma_R$, $Q_i$) is questionable, in the following of the study the 23 variables were calculated for each of the 545 rain events.

### 2.3 PCA analysis and normalization step

An important point to considerer is that most of these 23 variables do not allow the probabilistic assumptions associated with most exploratory statistical methods. They are often of great variability with highly skewed distributions and therefore are not normally distributed. Consequently the direct use of standard statistical methods on these data is made more complex and may lead to misleading interpretations (Daumas, 1982). An additional step is thus necessary in order to transform the original distributions to quasi normal distributed distributions. The type of normalizing transformation chosen for each variables was done empirically by testing 7 different possible transformations (Tab. 3). For each variable, the retained transformation is the one which gives the distribution the closest to a normal distribution, that is to say a kurtosis close to 3 and the skewness close to 0. For each indicator the selected transformation is given in Tab.2 last column.

**Table 3: Transformations used to normalize the variables described in Table 2**

After the normalisation step a Principal Component Analysis (PCA) was conducted on the learning data set. It follows that the two principal axes contain 73% of the total information while the first 5 principal axes are necessary to represent 90% of the total information. The $IET_p$ variable (#7) is very well correlated with axis 5 while others variables are not. This means that no linear relationship exist between $IET_p$ and the other variables. The correlation circle on axis 1 & 2 (Fig. 1.a.) shows that among the 23 variables 16 are well correlated with axis (close to unit circle) and are distributed more or less in 5 groups (Hereafter PCA groups). A first PCA group ($G_1$) can be identified by the variables which are grouped close to the first axis and are well correlated with it. This is for instance the case for the variables $\sigma_R$ (#9), $P_{C_N}$ (#17 – 18) and $\beta$ (#21 to 23). A second PCA group ($G_2$) composed of variables $R_{max}$ (#11) and $P_{C3}$ (#16) just above axis 1. The third PCA group ($G_3$) is formed only by the variable $P_{C2}$ (#15). The fourth PCA group ($G_4$) is composed of the variables $P_{c1}$ (#14) and $Ps,c$ (#20) and is well correlated with axis 2. The last PCA group ($G_5$) is formed by the variable $D_e$ (#1). The correlation circle on axis 1 & 3 (fig. 1.b.) shows that variables





$Q_1$ (#4), $Q_2$ (#5) and $M_0$ (#10) are quite well represented by these two axis. A similar remark can be made for variables $D_d$ (#3) and $\beta_{L1}$ (#21) on axis 1 & 4 (not shown in the paper).

Finally the PCA analysis clearly shows that within each PCA group many variables are highly correlated with each other, i.e. linearly dependant of each other's. This means that a number of variables can be removed without substantial loss of information. This leads to the following question: Which variables can be removed in order to get the most parsimonious subset of variables representative of the whole data set? PCA extracts summary variables which are linear combination of the original ones but do not allow for the selection of variables. To answer to this question, the approach we propose here is a method of global selection of variables which seek to identify the relevant variables in a dataset. It seems more interesting to select variables with a physical sense rather than using dimension reduction methods (e.g., the principal component analysis PCA which is more able for detecting linear relationships). The proposed method is derived from the genetic algorithms. The following paragraph provides some basics about genetic algorithms and presents how these can help us to make our selection.

**Figure 1: PCA on the training data set of the 23 variables described in Tab. 3. Left : Correlation circle on axis 1 & 2. Right : Correlation circle on axis 1 & 3. All the variables are normalised according to Tab. 2 last column.**

## 3. Variables selection with genetic algorithms

Genetic algorithms (GAs) (Holland, 1975) are stochastic optimization algorithms based on the mechanics of natural selection and genetics described by Charles Darwin. The principle is simple. We start at generation 1 with an initial population of potential solutions (chromosomes) arbitrarily chosen. In our case, each chromosome defines among the 23 variables a subset of variables which are potential candidates to describe the events. Then we evaluate the performance of each chromosome by the mean of a fitness function. The performance of a chromosome characterizes its ability to represent the topology of the whole dataset with a minimal number of variables. Based on this performance we create a new generation of chromosomes of potential solutions using classical evolutionary operators: selection, crossover and mutation. We repeat this cycle until the stop criteria is asserted.

### 3.1 Methodology

The GAs for variables selection are based on the following five steps (Fig.2.):

*Step 1- Initialization:* (Coding and initial population) we used binary coding as follows:

Let's define a chromosome $x = (x_1, x_2, ..., x_{23})$ as a vector in $\{0,1\}^{23}$ with:

$$\forall i \in \{1, ..., 23\}, \begin{cases} x_i = 1 & \text{The corresponding variable in Tab. 2 is selected} \\ x_i = 0 & \text{The corresponding variable in Tab. 2 is non selected} \end{cases} \quad (1)$$

Then we generated, randomly, a population $\{x^k, k=1, ..., 60\}$ of 60 chromosomes of dimension 23.

*Step 2- Evaluation:* For each of the $x^k$ chromosomes a Self-Organizing Maps $M(x^k)$ is associated. Self-Organizing Maps (SOM) introduced by Teuvo Kohonen (Kohonen, 1982) are a popular clustering and visualization algorithm (see section 4). The 234 rain events vectors belonging to the training data set are then used to train each of the 60 Maps. The maps $M(x^k)$ will only use for training the variables those the components in $x^k$ are equal to 1. Once training each of the 60 Maps $M(x^k)$ provides a topologic error $te(x^k)$ estimated on the whole (23) variables. A score $f(x^k)$ (fitness) is assigned to each chromosome $x^k$ as a





function of the quality of the SOM through its topology error and the number $nb$ of variables used to train the corresponding map. The analytical form of the fitness function is:

$$f(x^k) = \frac{1}{nb(x^k) \; te(x^k)}$$ (2)

With

$nb(x^k)$   :   *number of selected variables in chromosome $x^k$*
       $te(x^k)$   :       *topological error of the SOM $M(x^k)$*

This fitness function tends to provide low values when the number of selected variables or the topological error increases. The aim is to maximise the fitness function.

*Step 3-* Select the best chromosome x$^{Best}$ among the 60 chromosomes according to its fitness computed on the test dataset. If x$^{Best}$ remains the same during 50 generations then stop the procedure and select the relevant variables, i.e. the ones for which

the corresponding components are equal to 1 in x$^{Best}$. Else go to step 4.

*Step 4- Selection:* Create a new population of 60 chromosomes from the initial population by random sampling with replacement of chromosomes based on their probabilities calculated according to the formula:

$$\Pr(x^k) = \frac{f(x^k)}{\sum_{i=1}^{60} f(x^i)}$$ (3)

*Step 5- Reproduction:* Mutation and Crossover possibilities in the new population.

Mutation: It consists of modifying (or not) some components of the chromosomes. The probability of mutation is in general very low and is commonly set to $p = 10^{-7}$. In our case the number of necessaries generations to reach our objective is lower than a few hundred. Consequently in our case, the probability of a mutation is highly unlikely.

Crossover: First, we randomly draw $\frac{60}{2} = 30$ couples of chromosomes from our population. Then, for each couple ($x^k$, $x^l$) (called parents) one crossover point, noted $I_c$, is randomly drawn in the range [1, 23] using a discrete uniform law. Two new

chromosomes ($x^{k'}$, $x^{l'}$) are created in the following way:

$$\begin{cases} x^{k'} = (x_1^k, x_2^k, \ldots, x_{I_c}^k, x_{I_c+1}^l, x_{I_c+2}^l, \ldots, x_{23}^l) \\ x^{l'} = (x_1^l, x_2^l, \ldots, x_{I_c}^l, x_{I_c+1}^k, x_{I_c+2}^k, \ldots, x_{23}^k) \end{cases}$$ (4)

So from two parents we generate two children allowing having a new generation with the same number of chromosomes. Finally go to step 2.

**Figure 2: Diagram for the selection of variables based on a Genetic Algorithm associated with Kohonen Maps**

## 3. 2 Rain event parsimonious description

The genetic algorithm is applied to our datasets in order to obtain an optimal subset of variables forming a subspace who can be (in some sense) nearly informative of the global space with the particularity of non-redundancy of information. At the 187$^{th}$

generation we get a subspace formed by 5 variables that describe quite well the original space vectors, namely : Event duration $D_e$ (#1), Standard deviation $\sigma_R$ (#9), rain rate peak in event $R_{max}$ (#11), Rain event depth $R_d$ (#13), Absolute rain rate variation $P_{c1}$ (#14).





The 3 variables ($D_e$, $R_{max}$, $R_d$) selected with this data driven approach are commonly used in study of hydrological processes (Haile and al., 2011). We can note moreover that the commonly used variable $R_m$, which is simply calculated by dividing Rain event depth ($R_d$) by the duration ($D_e$), was not selected by the selection algorithm. This result is expected because it is correlated with the latter variables and since the algorithm provides a parsimonious description. Concerning the Absolute rain

rate variation ($P_{c1}$), the latter was proposed by Moussa and Bocquillon (1991), this variable tends to give information on the structure of the events and more specifically to smooth events with a small number of sharp peaks. Indeed this variable promotes low variations of $RR_t$ because $P_{C1}$ is in some sense a structure function of order $c_1$ of the variable $R_{max}$ (see #14 column 4 in Tab. 2) with a low value of the exponent ($c_1 = 0.5$). Finally the standard deviation variable ($\sigma_R$) which is a second order moment is the most common indicator to describe the variability of the precipitation rate within the rain event.

**4. SOM learned with the five selected variables**

The evaluation step is based on the ability of the selected variables to preserves as much as possible the topology of the initial space. This ability is quantified through the topologic error of the Self-Organizing Maps (Kohonen, 1982, 2001).

A SOM is a topological map composed of neurons. In our case, a neuron is a vector of dimension 23 containing the 23 variables defined previously. Each neuron has 6 neighboring neurons. SOM is an unsupervised neural network trained by a competitive

learning strategy that performs two tasks: vector quantization and vector projection. Different from K-means, SOM uses the neighborhood interaction set to learn the topological structure hidden in the data. In addition to the best matching referent vector, its neighbors on the map are updated, resulting in regions where neurons in the same neighborhood are very similar. It can be considered as an algorithm that maps a high dimensional data space to a two dimensional space called a map. A map can be used at the same time both to reduce the amount data by clustering and for projecting the data in a non-linearly way to

a regular grid (the map grid).

In this study a SOM with 8×8 = 64 neurons is considered. After training by the GA algorithm described in section 3 the map $M(x^{Best})$ can be used to affect to any event the best matching referent vector (neuron) according to the 5 variables previously selected. The obtained SOM $M(x^{Best})$ can be considered as an optimal representation of the initial data set.

Figure 3 shows the distances matrix. For each neuron the color indicates the mean distance between the neuron and its

neighbors. The value at the center of each neuron represents the number of rain events of the training data set captured by the corresponding neuron. All neurons capture rain events and a bit more than half of them capture between 3 and 5 rain events which is close to the value that would be obtained (234 / 64 ≅ 4) if the rain events were uniformly distributed on the map.

**Figure 3: Distance matrix of $M(x^{Best})$ map: The color of each neuron represents its distance with neighbouring neurons. Values inside each neuron provide the number of rain events of the training data set captured by the corresponding neuron. The black line separates the neurons in 2 classes using Hierarchical Ascendant Classification (see section 4.1). The arrows represent the gradients of variables $R_{max}$, $\sigma_R$ and $D_e$**

**4.1 Projection of the selected and unlearned variables on the SOM**

The five variables $D_e$, $\sigma_R$, $R_{max}$, $R_d$ and $P_{c1}$ used for training are referred to as 'selected' while the other 18 variables are referred to as 'unlearned'. In order to study the relation between variables figure 4 shows the projections for each variables of the $M(x^{Best})$ map obtained with the GA selection algorithm presented previously. The variables are discussed individually by considering their structuration but also by considering relationships between them. We note that the map is well structured for the majority of the variables. This good structuration of most of the variables confirms the ability of the selected variables to

summarize all the characteristics of rain events. Only few characteristics are not well represented. Note that almost all variables are structured according to the first or the second diagonal. Among them one can consider a first subset composed with





variables more or less structured according to the first diagonal. This is the case for the unlearned variable $D_d$ as well as the selected variables $P_{c1}$ and $D_e$. A second subset composed of variables more or less structured according to the second diagonal can be identified. This is the case for the unlearned variables $R_{m,r}$, $R_{m,}$, $P_{C_{Ni}}$ which are very similar to the selected variable $\sigma_R$. The unlearned variables $Q_3, P_{C3}, P_{S,C}$ belong also to the second subset and present a structure close to that of the selected

variable $R_{max}$.

The map can be related to the Principal Component Analysis conducted previously (Fig.1). As it can be shown in Fig. 4 the variables $P_{C3}$ (#16) and $R_{max}$ (#11) which have similar structure belongs also to the same PCA group, namely the group $G_1$ (see section 2.3, Fig. 1.a). It is interesting to note that the variables $P_{S,C}$ (#20) and $R_{max}$ (#11) which have also similar structure do not belong to the same PCA group (groups $G_4$ and $G_2$ respectively) and are uncorrelated (They are orthogonal in Fig.1a). This

remark means that the topological map reveal a relationship that cannot be detected with the PCA. As Rain event depth ($R_d$) depends both on the duration and intensity of the events, the corresponding map has a top-down structure. It appears clearly two distinct situations:

-        The events that bring the greatest amount of water (Fig. 4. brown neuron at the bottom right of $R_d$) are among the longest (see corresponding neuron of $D_e$) but do not have an extreme rain rate peak (see corresponding neuron of $R_{max}$) and are

quite smooth (see corresponding neuron of $P_{c1}$ and $\sigma_R$ ).

-        The events that bring a great amount of water (but less than previously) (Fig. 4. red neuron at the bottom left of $R_d$) have short durations (See corresponding neuron of $D_e$) but violent (see corresponding neuron of $R_{max}$) and are less smooth (see corresponding neuron of $P_{c1}$ and $\sigma_R$ ). This last case reflects situations of convective storm type.

The obtained map confirms the dependence structure of the two hydrological variables $R_d$ and $D_e$ studied in Gargouri and

Chebchoub (2010).

Variable $IET_p$ (Previous IET) : the map is not structured, reflecting the independence of the characteristics of a rain event with respect to the drought period preceding the event. This corroborates several previous works (Lavergnat and Gole, 1998, 2006; Akrour et al., 2015) relative to rain support simulation. These authors have noticed that successive rain and no rain periods are found to be uncorrelated, thus a rain time series can be considered by an alternation of rain event and no rain independently

drawn periods. That is similar to say that inter-event time (IET) doesn't characterize the rain events. Brown et al. (1983) also investigated a possible dependence between $IET_p$ and the intra-event characteristics and they conclude that the assessment of their data gave no indication that such dependency exists.

The $\beta_L$ variable: the $\beta_L$ variable (Llasat, 2001) is supposed to represent a measure of the convective nature of the rain, so it makes sense that the three variables $\beta_{L1}, \beta_{L2}, \beta_{L3}$ are structured similarly with the rain rate peak variable $R_{max}$. This relationship

clearly is visible on maps.

Several other relationships not detailed here are visible like the correlation between Normalized Absolute rain rate variation ($P_{C_{Ni}}$) and standard deviation of intensity ($\sigma_R$). We can conclude that the combination of the five selected variables summarizes information from rain events properties. The bad structuring for few variables is justified by the independence between these variables and the rain event properties; this is the case for the variable Dry Percentage in event $D_d$ or the variable $IET_P$.


**Figure 4: Projection of the $M(x^{Best})$ map according to the 23 variables. The red framed variables are those that were selected by the GA algorithm. The last two variables $D_m$ and $N_0^*$are defined later in section 5**





**4.2 Representation of rain events on SOM**

In order to provide additional information to validate the map we have compared both for the training and the test dataset each of the 23 variables with their corresponding value given by the SOM. For each of the 311 events of the test data set the best matching unit of the SOM, i.e. the neuron that is the closest of the event, is determined relatively to the five selected variables.

As an example Fig.5 shows for each event, the actual value of the unlearned variable $\beta_{L3}$ versus the corresponding value given by the best matching unit of the event. A spreading can be seen in particular in the central part while the spreading is relatively small for values located near the edges (and which are in largest number). The linear regression gives a quite good determination coefficient (R-square) (0.96 and 0.89 respectively for training and test data set). Table 4 gives the R-square for the 23 variables obtained on the training and test data set. As expected the coefficient of determination of the variable $IET_p$ is

very poor (*0.31/0.26*) since this variable is not related to the 5 selected variables and consequently cannot be well represented by the SOM (Fig. 4). The selected variables have good determination coefficients both on training and test dataset; this confirms the quality of learning and generalization ability of the SOM. The quality of the training step is confirmed by the fact that the R-square of the selected variables obtained on the test set are close to those obtained on the training set. The R-squares corresponding to the unlearned variables obtained on the training data set underline the ability of selected variables to provide

the information contained in the unlearned variables; on the test data set it denotes the ability of the SOM to deduce all events characteristics from only the selected variables.

**Figure 5: $\beta_{L3}$ variable versus its corresponding value given by the best matching unit: on training data set (circle) and on the test data set (star). The solid line correspond the first diagonal**

**Table 4: Coefficient of determination obtained on the training & test data sets. Values with dark grey background correspond to**
**the 5 selected variables**

**4.3 Hierarchical clustering of rain events**

We have seen that the distance matrix (Fig. 3) therefore confirms the successful deployment of the map. Based on the distance between neurons, it appears that neurons can be grouped to obtain a limited number of classes each with its own characteristics.

To group the 64 neurons in a few classes, a hierarchical cluster analysis was conducted (Everitt, 1974). Only the five selected variables were used for the classification and a Euclidian distance was selected for the hierarchical algorithm. Fig. 6 shows the obtained dendrogram applied to the 64 neurons.

**Figure 6  Dendrogram obtained from the Hierarchical Cluster Analysis of the 64 neurons of SOM**

In relation with physical processes involved, experts use to divide the rain events into two classes: stratiform and convective events. Although this classification is quite crude because stratiform and convective events can sometime exist inside the same rain event such a classification is very used. Concerning times series, authors use very simple scheme to separate stratiform and convective rain types. For simplicity reason, rain classification is sometimes defined using instantaneous rain rate and

standard deviation estimated over consecutive samples. As an example Bringi and al. (2003) defined stratiform rain samples when standard deviation of rain rate over five consecutive 2-min samples is less than 1.5 mm.h$^{-1}$ and convective rain samples are defined for rain rate greater or equal to 5 mm.h$^{-1}$ and standard deviation of rain rate over five consecutive 2-min samples greater than 1.5 mm.h$^{-1}$.

Firstly we cut the dendrogram in two classes. The first class is composed of 51 neurons and contains 79% of the observations



while the second one is composed of 13 neurons and contains 21% of the observations. The solid black line in Fig. 3 gives the frontier between these two classes. The first class which is composed by the larger number of neurons is characterized in the majority of cases by relatively low rain rates. This can be seen by examining the structure of the map according to the mean rain rate variable ($R_m$). Moreover the analysis of standard deviation (small values of $\sigma_R$), absolute rain rates $P_c$ (high values of

$P_{c1}$ and low values *of* $P_{c3}$) shows that this class is more or less characterised by quiet and homogeneous events. The analysis of event duration ($D_e$) shows that this class contains both short and long duration but with a preponderance for the latter. These characterizations correspond quite well to description of stratiform and stable precipitations, which are often due to a slow and large-scale uprising of a mass of moist air that condenses evenly.

The second group is characterized by a smaller number of neurons. It corresponds to the higher values of the mean rain rates

($R_m$) and rain rate peaks ($R_{max}$). The variables $\sigma_R$, $P_c$ have the opposite values compared to those of the previous group. Most of the event durations ($D_e$) of this group are short except the neurone #64 (right bottom on the maps). This group fits well with the definition of convective events which result from the rapid rise of air masses loaded with moisture, for buoyancy. This convection moist air can cause the development of cumulus with vertical extensions that can exceed 10 km altitude and leading to heavy rain.

The analysis of the structure of variables $\beta_{L1}, \beta_{L2}, \beta_{L3}$ in Fig. 4 comforts the previous interpretation of the two groups. These three variables, which are representative of convective rains, have high values for neurons belonging to this group.

Figures 7.a and 7.b show the neurons in the $R_m$, $\beta_{L3}$ and $P_{C2}$ subspace. We can note that these 3 variables were not used in the learning step. In spite of that, the two classes are well separated even if an overlap can be seen in Fig 7.a due to the neurons #64 (bottom right on the map). As it will be seen further although it belongs to convective class, it nevertheless has some

characteristics of the stratiform class.

**Figure 7: Representation of the neurons in the subspaces Rm, $\beta_{L3}$ and Rm, $P_{C2}$. Stars represent neurons belonging to group one (stratiform) and square to neurons of group 2 (convective)**

We conclude that this unsupervised automatic clustering based on the five selected variables allow to implement properly the well-known 2 classes (stratiform and convective) classification. Note that this classification unlike those found in the literature was done with no a priori since it results from an unsupervised process.

**4.4 Classification of events into several classes**

From the stratiform and convective classification described above it is interesting to refine the two classes in subclasses. For example, the synoptic precipitations caused by the mid-latitude depressions are an example of stratiform precipitations. They manifest themselves in the body rainy disturbances associated with warm and cold fronts. The very low light rains (the drizzle) caused by stratus or stratocumulus are also part of the range of stratiform precipitations. They occur either in anticyclonic conditions or in the warm sector of a disturbance. The associate rain depths ($R_d$) are minimal, and they usually have no

hydrological impact other than the superficial wetting. To identify subclasses we had to refine the classification into a number of unknown subclasses $n > 2$.

An important step of hierarchical clustering is the selection of the optimal number of partitions ($n_{opt}$) of the data set (Grazioli et al., 2015). Many indices can be employed to evaluate each partition from the point of view of data similarity only. Most of these indices evaluate the scattering inside each cluster with respect to the distance between clusters and they assign relatively

better scores to partitions with compact and well-separated clusters. We have tested different indices but they do not provide the same number of subclasses (between 2 and 32 with the indices we have tested). We can notice that they do not consider





the physical sense of each class. Finally we choose $n_{opt} = 5$ because for higher values we get classes with the same physical sense which means a useless classification. The new classification using fives subclasses is presented in Fig. 8.

**Figure 8: Hierarchical Clustering of the map in five subclasses. Colours represent the subclass numbers: Subclass 1 : Dark blue, Subclass 2 : blue, Subclass 3 : Green, Subclass 4 : Orange, Subclass 5 : Red**

Finally, among the five subclasses, two are part of the stratiform class and the three others are part of the convective class. On the training dataset, the first subclass represents 12% of the whole events and respectively 68%, 1.2%, 6.8% and 12% for the

subclasses 2 to 5. The characteristics of these five subclasses are summarized below and in Tab. 5. The five selected variables are remarkably heterogeneous between classes meaning the accuracy of these variables for clustering:

**Subclass 1** (drizzle and very light rain):  the main feature of this class is the very low mean ($R_m$) and standard deviation $\sigma_R$ of event rain rates in addition to the features of the superclass. The mean event rain rates are in the range [0, 0.5] mm.h$^{-1}$ with a

mean value of 0.36 mm.h$^{-1}$ and $\sigma_R$ in the range [0, 3] mm.h$^{-1}$ with a mean value of 0.1 mm.h$^{-1}$. This class give a small amount of water although the event duration is high. This subclass of events is characteristic of drizzle. We can note also that low value of $\beta_{L3}$ is a good indicator (< 0.01) for drizzle.

**Subclass 2** (The "normal" events): a relatively broad class which contain 68 % of the whole events with mean event rain rate ($R_m$) in the range [0.5 , 6] mm.h$^{-1}$ and a mean value of 1.48 mm.h$^{-1}$. The standard deviation $\sigma_R$ is in the range [1, 10] mm.h$^{-1}$

with a mean value of 2. This subclass is characterised by a quite important relative variation of some parameters ($D_e$, $R_m$, $P_{c1}$ for instance) and dry periods ($D_d$) which could be long enough.

The three remaining subclasses compose the convective class events. They are characterized by a strong temporal heterogeneity and significant intensities. This convective class is divided according to the rain event depth into:

**Subclass 3** contains relatively long events ($D_e$) with high values of the rain event depth ($R_d$) variable and $P_{c1}$. This class

represents events with a very small occurrence (1.2%).

**Subclass 4** contains relatively short events ($D_e$) with rain rate peak that exceeds $R_{max} > 50$ mm.h$^{-1}$ in addition to strong heterogeneity ($\sigma_R$, $P_{C2}$ and $P_{C3}$ are high) and large values of the convective indicator ($\beta_{L3}$).

**Subclass 5**, the events of this subclass display relatively low values of the rain event depth ($R_d$) variable. It is due to the short duration of the events ($D_e$). The variables $\sigma_R$ and $P_{C3}$ remain high. Another feature of this subclass is that it is composed

entirely of continuous events without embedded small dry periods (low values of $D_d$ in Fig.4 and Tab. 5).

**Table 5: Summary of rain events subclasses calculated on the training dataset.**

To conclude this section, this new classification allows refining the traditional stratiform – convective classification in five homogeneous sub classes but sufficiently heterogeneous between them. The last step of this study will consist to assess if the homogeneity character of each class is preserved at the microphysics scale and glimpse the possible existence of relationships between microphysics and macrophysics (hydrological information) scales.

**5. Microphysics point of view**

The study of the microphysical properties of rain is based on the drop size distribution $N(D)$ which is the number of raindrops



per unit volume and per interval of diameter *D*. The shape of the drop size distribution *N(D)* reflects the microphysical processes involved. The identification of both the drop size distribution features and the precipitation types is useful and important for numerous applications. For example they are used in the calculation of heating profiles in the precipitation parameterization for atmospheric models or in the understanding of microphysical processes, as well as in the development of

rain retrieval algorithm for remote sensing measurement. Microphysical characteristics act as hidden variable impacting the relationship between microwave remote sensing measurement of and rain water quantity (Ulaby, 1981; Iguchi, 2009). Obtaining microphysical characteristics of rainfall events from conventional rain gauges could greatly help to improve active or passive remote sensing of rain, in particular spatial one.

A general expression of the drop size distribution defined by Testud et al. (2001) is commonly used. It allows distinguishing

a stable shape function *f* and the variability induced by rain. This variability is represented by two microphysics parameters namely the mean volume diameter ($D_m$) and the parameter $N_0^*$. In some of the references $N_w$ is used instead of $N_0^*$ but the various references do not use the exactly the same units, in particular Bringi et al. (2003) and Suh et al. (2016) use mm$^{-1}$m$^{-3}$ for $N_w$ units instead of mm$^{-4}$ for $N_0^*$.

$$N(D) = N_0^* f\left(\frac{D}{D_m}\right) \qquad \text{[m}^{-4}\text{]} \qquad (5)$$

Their definition is recalled below:

$$D_m = \frac{M_4}{M_3} \ \text{[mm]}, \quad N_0^* = \frac{4^4}{\Gamma(4)}\frac{M_3^5}{M_4^4} \quad \text{[m}^{-4}\text{]} \qquad (6)$$

Where $M_i$ is *i*th-order moment of the drop size distribution *N(D)*

$$M_i = \int_0^{+\infty} N(D)D^i dD \qquad (7)$$

Usually rain samples are analysed by computing the microphysical parameters ($D_m$ and $N_0^*$) for each rain samples obtained

for a given time scale. In this study the drop size distribution *N(D)* is obtained by considering the whole raindrop collection corresponding to each rain event of (variable) duration $D_e$. It thus leads to one couple ($D_m, N_0^*$) of microphysics variables per rain event where other authors rely on values computed over a fixed time scale.

Projections of the trained map according to $D_m$ and $N_0^*$ are shown in Fig.4 (bottom right). The first remark is that the two maps are well structured. We see that the two parameters vary oppositely. Although these parameters were not learned directly we

notice that the relationship between the two microphysical parameters is clearly taken into account by the information used to structure the map (the 5 selected variables). Moreover, the existence of relations between microphysical and macro physical features is also confirmed in Fig. 4, in which the macro physical variables $\sigma_R$ and $R_{max}$, used to learn the SOM, both display patterns similar to these found on the $D_m$ map.

Many authors try to associate specific microphysical properties to each precipitation types (convective or stratiform) (among

others, Atlas et al., 1999; Bringi et al., 2003; Marzuki et al., 2013; Suh et al. 2016). Considering Fig. 4 and the convective/stratiform classification of the section 4.3, we can confirm that precipitation events classified as stratiform express small values for $D_m$ and large values for $N_0^*$. For the convective class we notice the opposite (i.e. larger values for $D_m$ and smaller values for $N_0^*$). Similar observations are also reported by Testud et al. (2001). It can also be noticed that the two microphysics variables are more homogeneous in the convective class while the stratiform class shows larger variability.

To better study the microphysical information embedded in our data set we are going to study the relationship between the two microphysical parameters through the referent vectors (the neurons) of the map, which encompass the information of the original rain events.





Figure 9 shows the variable $D_m$ versus $N_0^*$ for the 64 neurons of the map. To better discuss the microphysics related to the two rain types (stratiform and convective), the relationship is displayed using distinct markers to identify the five subclasses defined in section 4.4. The two solid lines show the linear regressions calculated on stratiform and convective classes.

For stratiform subclasses (1 and 2) one can see a clear relationship between the two variables. The microphysics characteristics of these two subclasses are clearly distinct. Indeed, the subclass 1 (drizzle and light rain) has the smallest $D_m$ and the highest $N_0^*$ and have small ranges of variation. On the contrary like for macro physics variables (see section 4.4) the microphysics characteristics of the subclass 2 (normal events) is much more heterogeneous. The knowledge of $D_m$ allows to easily discriminate the corresponding subclass. Hence an event with values of $D_m$ in the range [0.5, 1] millimetre belongs to the subclass 1. In the same way, an event with values of $D_m$ in the range [1, 1.7] millimetre belongs very probably to subclass 2.

For convective events (subclasses 3, 4, 5) little differences can be noticed considering $N_0^*$. In the range [1.7 , 2.5] mm two neurons belonging to subclass 4 are close to a neuron belonging to subclass 5 and therefore have similar microphysics. One can note also the presence of three isolated neurons belonging to subclass 2 (stratiform) even if they are located far from other neurons of subclass 2. They are characterised by quite strong values of $D_m$ (2 mm) and low values of $N_0^*$. We checked that the corresponding events are a mixture of stratiform and convective rain. A typical case consists of convective rain at the beginning
of the event with strong rain rates while the rest of the event is stratiform with low rain rates values and small variations.

**Figure 9:** *Microphysical variable $N_0^*$ versus $D_m$ for the five rainy event subclasses. The three neurons corresponding to mixed events are circled. Dashed lines correspond to borders $D_m > 1.66$ and $Log(N_0^*) > 6.15$*

According to our classification, Fig.9 allows to conclude that there exist some relationships between macro physical and microphysical variables. Nevertheless the knowledge of the couple $(D_m, N_0^*)$ does not allow determining in some cases the correct subclass.

The authors who are concerned by the identification of microphysical features with precipitation type use simple schemes
based on rain rate estimated for a fixed integration time (a few minutes) to separate stratiform and convective rain type. They also use these simple schemes to label $D_m$ and $N_0^*$ as stratiform or convective (Testud et al., 2001). This approach is significantly different from ours that assumes that all samples of an event belong to the same class. Our $N_0^*$ and $D_m$ are thus not computed on a fixed period but rather on the time scale of a given event. Consequently it has to be noticed that the $N_0^*$ found in this study do not cover the same range as the ones of the previously cited studies even if there is a good agreement concerning the $D_m$
range.

Many previous authors observed that the drop size distribution is closely linked to processes that control rainfall development mechanisms. In the case of stratiform rainfall, the residence time of drops is relatively long and raindrops grow by the accretion mechanism. In convective rainfall raindrops grow by the collision–coalescence mechanism associated with relatively strong vertical wind speeds. Numerous studies exist concerning the variability of $N_0^*$ and $D_m$, Bringi et al. (2003) study rain samples
from diverse climates and analyse their variability in stratiform and convective rains while Marzuki et al. (2013) investigate variability of raindrop size distribution through a network of Parsivel disdrometers in Indonesia and Suh et al. (2016) investigate raindrop size distribution in Korea using a POSS disdrometer. For stratiform rain they all observe that $N_0^*$ and $D_m$ are nearly log-linearly related with negative slope. This is consistent with what can be seen in Fig. 9 for the two stratiform subclasses (1 and 2). Even the three distinct neurons isolated from the others seem to be ruled by the same relationship.

During convective rain, Marzuki et al. (2013) note that the increase of $N_0^*$ with decreasing $D_m$ is also close to log-linear with flatter slope. In our case the dependence is also log-linear with a slope slightly flatter for convective events than for stratiform ones. In the works previously cited the data are aggregated over time by campaign or by site on the basis of criteria computed



over a fixed time period. We believe that this process is not well appropriate to display the properties of convective events given their strong variability and their shorter characteristic time. In our study we are able to retrieve the log-linear relationship between $N_0^*$ and $D_m$ without directly learning it.

Using our algorithm on various macroscopic properties by rain event we also take into account the variability of rain within a

rain event. Fig 9 shows clearly that the spreading of parameters $N_0^*$ and $D_m$ inside each subclass has the same magnitude than the distance between subclasses. This remark confirms the hypothesis of Tapiador et al. (2010) : the intra event variability range can exceed the inter event variability due to events from different precipitation systems. It is thus best to examine more generally the properties of events instead of studying the distinction between stratiform and convective processes through individual samples. The three isolated neurons of subclass 2 (circled in Fig. 9) previously mentioned share the properties of

the other events of their subclass (i.e. same slope for the log-linear relationship between $N_0^*$ and $D_m$). This example confirms the ability of the methodology to keep the pertinent macroscopic information to cluster rain events that allows restoring intra event variability as well as microphysical information.

In their study, Suh et al. (2016) also compare $\log(N_0^*)$ and $D_m$ pdf for stratiform and convective samples over a 4-year period. Based on the $D_m$ pdf of both stratiform and convective classes they compute a threshold value of $D_m$. For values of $D_m > 1.66$

mm the rainfall samples are mostly convective. They are mostly stratiform otherwise. This finding is also consistent with the results of Atlas et al. (1999) who also found a threshold $D_m$ value distinguishing convective and stratiform rainfall. In Fig. 9, it can be seen that this threshold is confirmed (vertical solid line), $D_m$ smaller than 1.6 mm correspond to stratiform events while values higher are mostly convective. As we consider events, we also have the three neurons corresponding to a "mixed event" beyond this threshold.

Concerning $N_0^*$ Suh et al. (2016) in their Fig. 4c one can observe that the pdf for convective rainfall is higher than that of stratiform for $\log(N_0^*) > 6.2$ ($N_w = 3.2$ in their figure). As previously told, by considering events, our range of values for $N_0^*$ is smaller from the ones displayed in the other studies. In our study all the neurons labelled as convective have $\log(N_0^*) < 6.15$ which is quite close from the value 6.2 found in Suh et al. (2016).

Given the overall good retrieval of microphysical information, from macrophysical parameters we can consider that the

topological map successfully stored some implicit information embedded in the data set. Since macrophysical parameter values are ruled by the microphysical properties of rain we can wonder how this is done. First of all, the selection of the map gathers events similar from each other while making sure, through the minimization of the topological error, that the unfolding of the map is correct. Hence a neuron is closer to his neighbour than to any other neuron of the map. This criterion insures, as much as it can be done, a partition of the data space in connected subparts. Hence the neurons of the map could be linked with the

underlying processes ruling rainfall.

6. **Conclusion**

Even if the definition of 'rain event' is relatively subjective, this study underlines the benefit of event analysis instead of samples analysis. This data driven analysis of events shows that rain events exhibit coherent features. In fact, the discrete and

intermittent natures of rain processes make the definition of some features inadequate when defined on a fixed duration. A too long integration times (hour, day) lead to mix observations that correspond to distinct physical processes and also to mix rainy and clear air periods in the same sample. A too small integration time (seconds, minutes) will lead to noisy data with a great sensibility to detector characteristics (capture area, detection threshold and noise). The analysis on the whole rain event instead of individual short duration samples of a fixed duration allows to clarify relations between features, in particular between

macro physical and microphysical ones. This approach allows suppressing the intra-event variability partly due to measurement uncertainties and allows focusing on physical processes.

Once clarified the definition of an event it is to select a small number of relevant variables to describe it. A new data-driven approach is developed to select the relevant variables. This approach present generic properties and can be adapted to many





multivariate applications. A genetic algorithm associated to Self-Organizing Map (SOM) clustering allow to select in an unsupervised way an optimal subset of five macro physical variables in minimizing a score function depending both of the topology error of the SOM and the number of used variables. This score provides a parsimonious description while preserving as much as possible the topology of the initial space.

In the context of rain time series studies and due to the wide variety of center of interest (hydrology, meteorology, climate, forecast) numerous variables (based on rain rates records) are used to describe precipitation records. The proposed algorithm gets a subspace formed by 5 among the 23 features found in the literature. We show that these five features selected (in an unsupervised way) by the algorithm describe the main characteristics of rainfall events from a macro physic point of view. These features are: the event duration, the rain rate peak, the rain event depth, the event rain rate standard deviation and the
absolute rain rate variation of order 0.5.

    To confirm relevance of the five selected features the corresponding SOM is analyzed. This analysis shows clearly the existence of relationships between features. It also shows the independence of the inter-event time ($IET_p$) feature or the weak dependence of the Dry percentage in event ($D_{d\%e}$) feature. This confirms that a rain time series can be considered by an alternation of independent rain event and no rain period. A hierarchical clustering is performed. The well-known division
between stratiform and convective events appears clearly. This classification into two classes is then refined in 5 fairly homogeneous subclasses. The stratiform class was divided into 2 subclasses: a drizzle / very light rain subclass and a normal events subclass. The convective class was divided into 3 subclasses characterized by a strong temporal heterogeneity and significant rates.

The data driven analysis performed on whole rain events instead of fixed length samples is relevant to study relations between
macrophysics (based on rain rate) and microphysics (based on raindrops) features. Strong relationships between microphysical and macro physical characteristics were identified. We show that among the 5 identified subclasses some of them have specific microphysics characteristics.

The relation between microphysical and macro physical characteristics can suggests many implications especially for remote sensing. In the context of weather radar applications the microphysical rain properties are needed to estimate rain rates through
the Z–R relations. The estimation of microphysical characteristics from easily observable by rain gauges features can plays an important role in development of the quantitative precipitation estimation (QPE).

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





**Tables**

|  | Observation period | Availability (%) | Number of rain event |
|---|---|---|---|
| Learning data set | 01/01/2013-12/31/2014 | 96.4% | 234 |
| Test data set | 04/16/2008-01/31/2012 | 60% | 311 |

**Table 1: Observation periods and availability of DBS observations and number of rain events for learning and test data sets**





| Number (#) | Feature name | symbol | Formula | Normalisation |
|---|---|---|---|---|
| 1 | Event duration | $D_e$ | $D_e = T_{end} - T_{begin} + 1$ [min] <br> With $T_{begin}$: Event start time and $T_{end}$: Event end time | 1 |
| 2 | Mean event rain rate | $R_m$ | $R_m = \frac{1}{D_e}\sum_{t=T_{begin}}^{t=T_{end}} RR_t$ [mm h⁻¹] | 2 |
| 3 | Intra-dry duration | $D_d$ | $D_d = \sum_{t=T_{begin}}^{t=T_{end}} I_t$ [min] <br> With $I_t = \begin{cases} 1 & if\ RR_t = 0\ [mm\ h^{-1}] \\ 0 & else \end{cases}$ | 0 |
| 4 | First quartile | $Q_1$ | The 25th percentile [mm h⁻¹] | 0 |
| 5 | Median | $Q_2$ | the 50th percentile [mm h⁻¹] | 0 |
| 6 | Third quartile | $Q_3$ | The 75th percentile [mm h⁻¹] | 2 |
| 7 | Previous IET | $IET_p$ | $IET_p = T_{begin}(current\ event) - T_{end}(previous\ event) + 1$ [min] | None |
| 8 | Mean rain rate over the rainy period | $R_{m,r}$ | $R_{m,r} = \frac{1}{(D_e - D_d)}\sum_{t=T_{begin}}^{t=T_{end}} RR_t$ [mm h⁻¹] | 3 |
| 9 | Event Rain rate std. | $\sigma_R$ | $\sigma_R = \sqrt{\frac{1}{D_e}\sum_{t=T_{begin}}^{t=T_{end}}(RR_t - R_m)^2}$ [mm h⁻¹] | 2 |
| 10 | Mode | $M_0$ | $M_0$ = the most frequent $RR_t$ | 0 |
| 11 | Rain rate peak | $R_{max}$ | $R_{max} = \max(RR_t)$ | 2 |
| 12 | Dry Percentage in event | $D_{d\%e}$ | $D_{d\%e} = \frac{D_d}{D_e}$ | 5 |
| 13 | Rain event depth | $R_d$ | $R_d = R_m * D_e / 60$ [mm] | 0 |
| 14 | Absolute rain rate variation of order c | $P_{c1}$ | $P_{c_i} = \sum_{t=T_{begin}}^{t=T_{end}-1} |RR_{t+1} - RR_t|^{c_i}$ <br> For $c_i = 0.5, 1, 2$ | 6 |
| 15 | | $P_{c2}$ | | 3 |
| 16 | | $P_{c3}$ | | 2 |
| 17 | Normalized Absolute rain rate variation of order $c_i$ | $P_{C_{N1}}$ | $P_{C_{Ni}} = \frac{P_{c_i}}{D_e}$ <br> For $i = 1..3$ | 3 |
| 18 | | $P_{C_{N2}}$ | | 2 |
| 19 | | $P_{C_{N3}}$ | | 0 |
| 20 | Absolute rain rate variation of order C and threshold S | $P_{S,C}$ | $P_{S,C} = \sum_{t=T_{begin}}^{t=T_{end}-1} |\max[(RR_{t+1} - S), 0] - \max[(RR_t - S), 0]|^C$ <br> With $s = 0.3$ and $c = 2$ | 6 |
| 21 | $\beta_L$ parameter | $\beta_{L1}$ | $\beta_{L_i} = \frac{\sum_{i=T_{begin}}^{T_{end}} RR_t\ \theta(RR_t - L_i)}{\sum_{i=T_{begin}}^{T_{end}} RR_t}$ | 5 |
| | | $\beta_{L2}$ | For $L_i = 0.3, 1, 3\ mm\ h^{-1}$ | 0 |
| 22 | | | With $\theta(RR_t - L_i)$ is the Heaviside function defined as | |
| 23 | | $\beta_{L3}$ | $\theta(RR_t - L_i) = 1\ if\ RR_t \geq L_i$ <br> $\theta(RR_t - L_i) = 0\ if\ RR_t < L_i$ | 0 |

**Table 2: The 23 variables identified in the literature to characterize a rain event**





| Transformation number | Transformation name | Formula f(x) | Note & remark |
|---|---|---|---|
| 0 | Standardisation | $\dfrac{x - mean(x)}{std(x)}$ | - |
| 1 | Power | $x^n$ | $n = 0.05$ |
| 2 | Boxcox | $\dfrac{(x^\gamma - 1)}{\gamma}$ | $\gamma = -0.1$ |
| 3 | | | $\gamma = -0.2$ |
| 4 | | | $\gamma = -0.3$ |
| 5 | Arc-sin of square | $\arcsin\sqrt{x}$ | Data are between 0 and 1 |
| 6 | decimal Logarithm | $Log(x + c)$ | $c = 0.1$ |

**Table 3: Transformations used to normalize the variables described in Table 2.**

| Variables | $D_e$ | $R_m$ | $D_d$ | $Q_1$ | $Q_2$ | $Q_3$ | $IET_p$ | $R_{m,r}$ | $\sigma_R$ | $M_0$ | $R_{max}$ | $D_{d\%e}$ |
|---|---|---|---|---|---|---|---|---|---|---|---|---|
| $R^2$ training data set | 0.96 | 0.91 | 0.58 | 0.57 | 0.48 | 0.77 | 0.31 | 0.93 | 0.97 | 0.50 | 0.96 | 0.52 |
| $R^2$ Test data set | 0.93 | 0.84 | 0.55 | 0.57 | 0.50 | 0.74 | 0.26 | 0.84 | 0.86 | 0.54 | 0.82 | 0.50 |

| Variables | $R_d$ | $P_{c1}$ | $P_{c2}$ | $P_{c3}$ | $P_{cN1}$ | $P_{cN2}$ | $P_{cN3}$ | $P_{S,C}$ | $\beta_{L1}$ | $\beta_{L2}$ | $\beta_{L3}$ | - |
|---|---|---|---|---|---|---|---|---|---|---|---|---|
| $R^2$ training data set | 0.97 | 0.97 | 0.93 | 0.94 | 0.91 | 0.95 | 0.78 | 0.94 | 0.70 | 0.89 | 0.96 | |
| $R^2$ Test data set | 0.94 | 0.99 | 0.91 | 0.83 | 0.83 | 0.85 | 0.71 | 0.82 | 0.61 | 0.76 | 0.89 | - |

**Table 4: Coefficient of determination obtained on the training & test data sets. Values with dark grey background correspond to the 5 selected variables**

| | Stratiform events | | Convective events | | |
|---|---|---|---|---|---|
| | Subclass 1 | Subclass 2 | Subclass 3 | Subclass 4 | Subclass 5 |
| Variables | Mean | Mean | mean | mean | mean |
| $D_e (min)$ | 321 | 149 | 464 | 75 | 49 |
| $\sigma_R$ | 0.36 | 2.01 | 3.62 | 11.7 | 9.64 |
| $R_{max}(mm\ h^{-1})$ | 2.08 | 10 | 22 | 52.7 | 36.06 |
| $R_d(mm)$ | 1.99 | 2.62 | 11.24 | 6.9 | 2.72 |
| $P_{c1}$ | 75.7 | 64.5 | 193 | 78.2 | 40.94 |
| $R_m(mm\ h^{-1})$ | 0.37 | 1.48 | 2.35 | 7.85 | 7.11 |
| $D_d(min)$ | 80 | 31 | 75 | 11 | 1 |
| $\beta_{L3}$ | 0.01 | 0.42 | 0.48 | 0.89 | 0.86 |

**Table 5: Summary of rain events subclasses calculated on the training dataset.**



# Figures

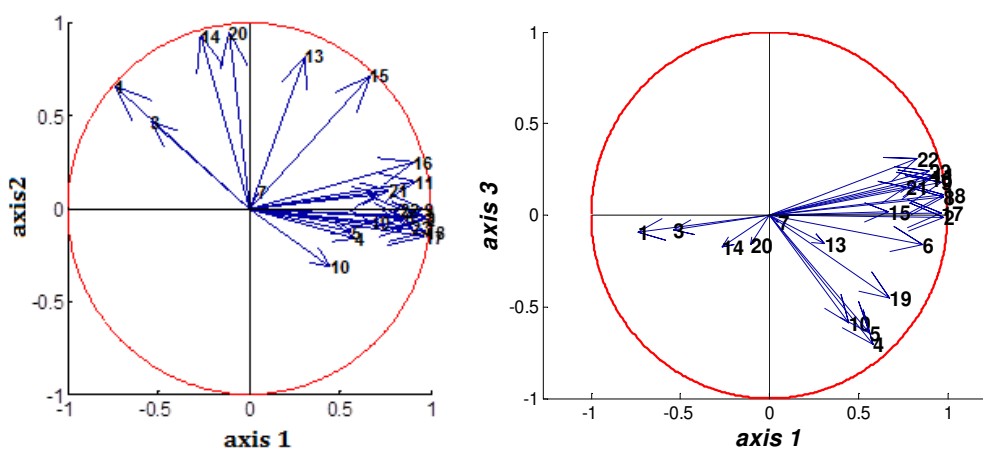

5    **Figure 1:** **PCA on the training data set of the 23 variables described in Table 3. Left : Correlation circle on axis 1 & 2. Right : Correlation circle on axis 1 & 3. All the variables are normalised according to Table 2 last column.**



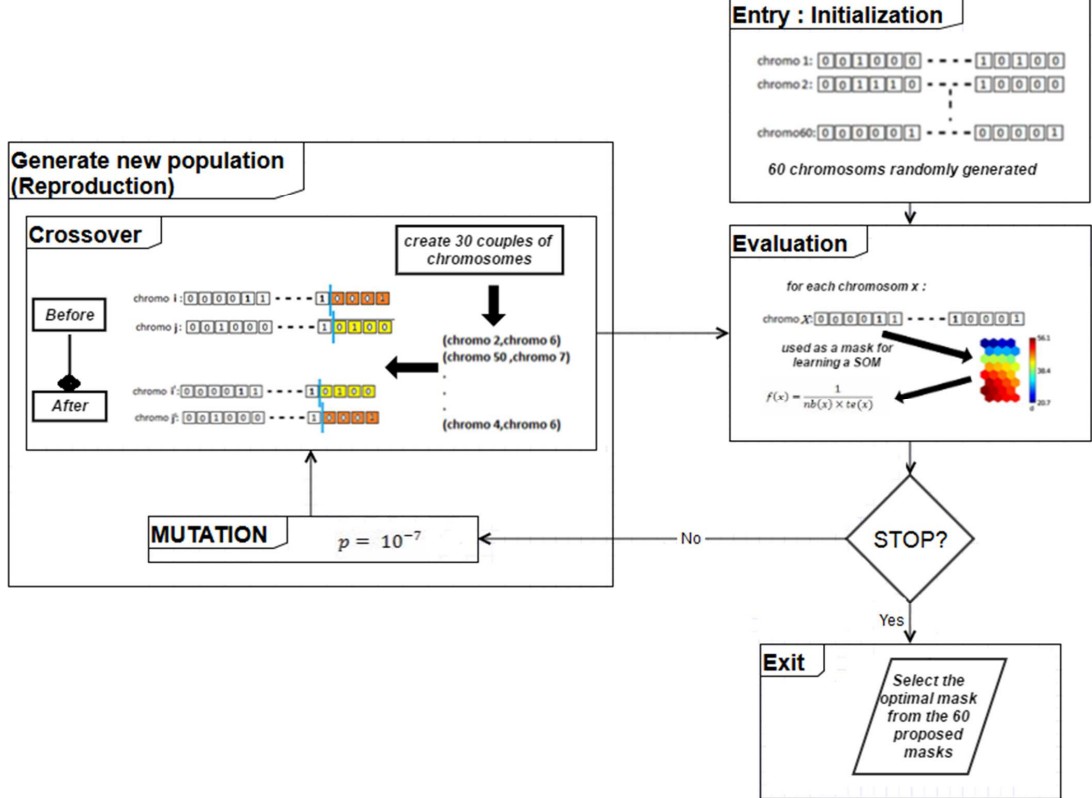

5     **Figure 2: Diagram for the selection of variables based on a Genetic Algorithm associated with Kohonen Maps**





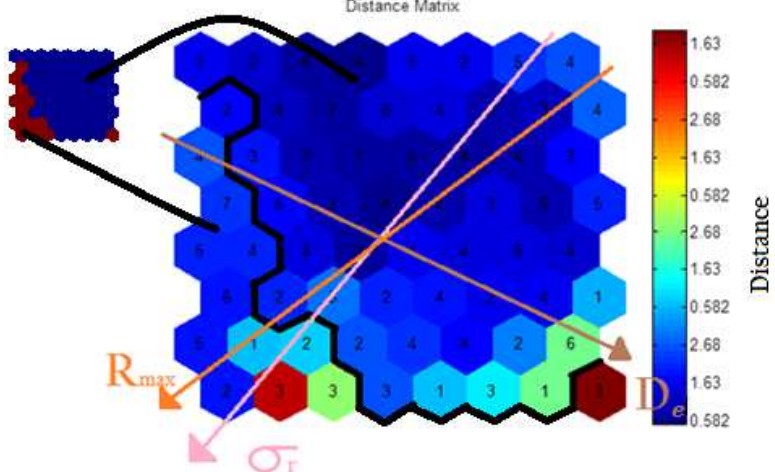

**Figure 3: Distance matrix of M(x$^{\text{Best}}$) map: The color of each neuron represents its distance with neighbouring neurons. Values inside each neuron provide the number of rain events of the training data set captured by the corresponding neuron. The black line separates the neurons in 2 classes using Hierarchical Ascendant Classification (see section 4.1). The arrows represent the gradients of variables R$_{\text{max}}$, $\sigma_R$ and D$_e$**





**Figure 4: Projection of the M(x^Best) map according to the 23 variables. The red framed variables are those that were selected by the GA algorithm. The last two variables $D_m$ and $N_0^*$ are defined later in section 5**





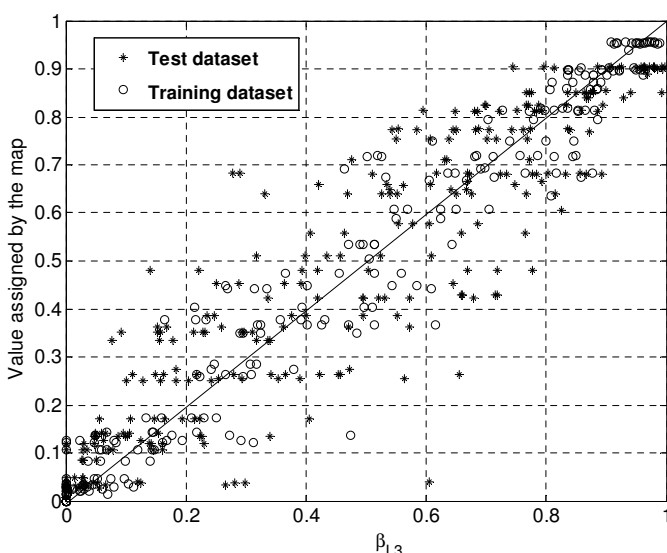

**Figure 5: $\beta_{L3}$ variable versus its corresponding value given by the best matching unit: on training data set (circle) and on the test data set (star). The solid line correspond the first diagonal**





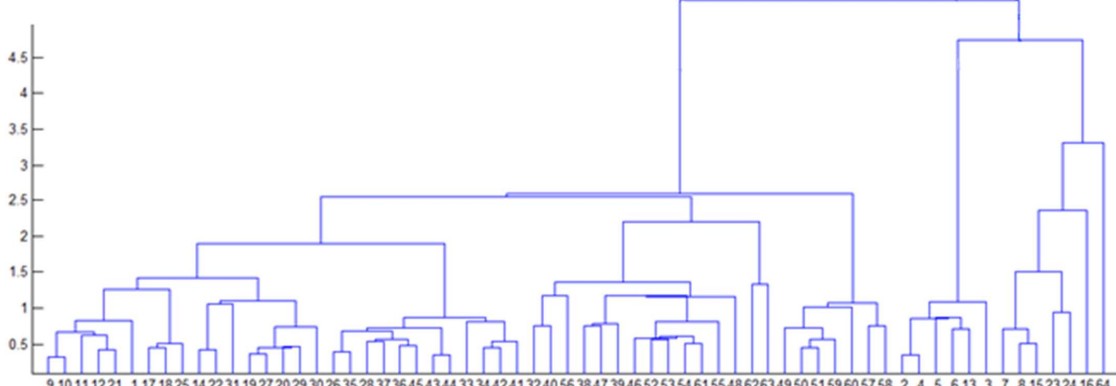

**Figure 6 Dendrogram obtained from the Hierarchical Cluster Analysis of the 64 neurons of SOM**





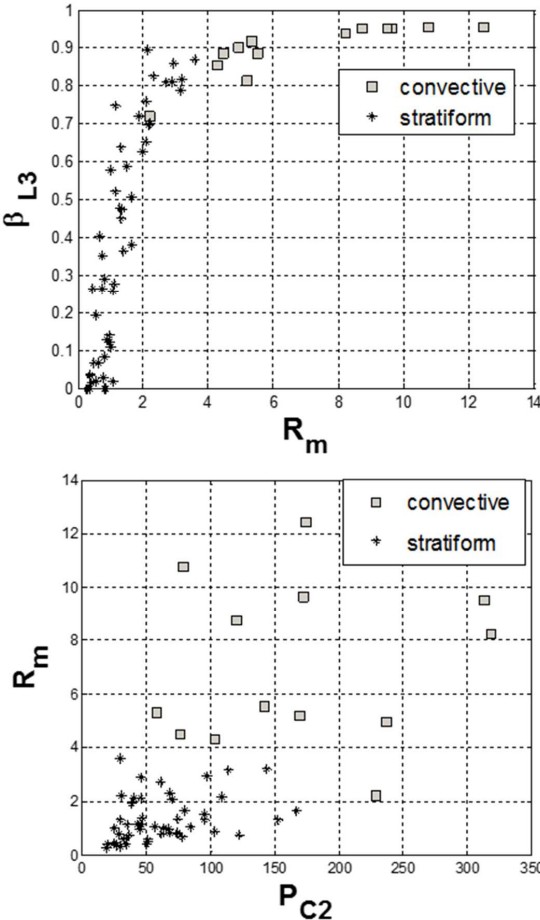

5    Figure 7: Representation of the neurons in the subspaces Rm, $\beta_{L3}$ and Rm, $P_{C2}$. Stars represent neurons belonging to group one
(stratiform) and square to neurons of group 2 (convective)





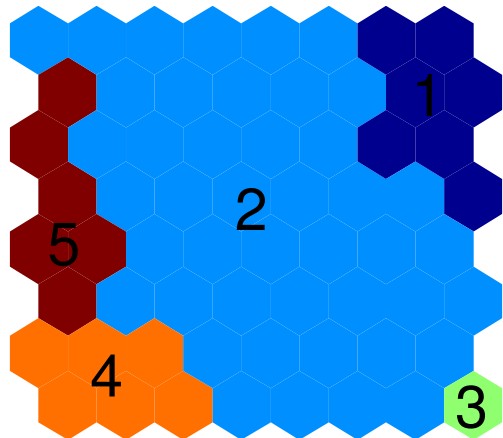

**Figure 8: Hierarchical Clustering of the map in five subclasses. Colours represent the subclass numbers: Subclass 1 : Dark blue, Subclass 2 : blue, Subclass 3 : Green, Subclass 4 : Orange, Subclass 5 : Red**



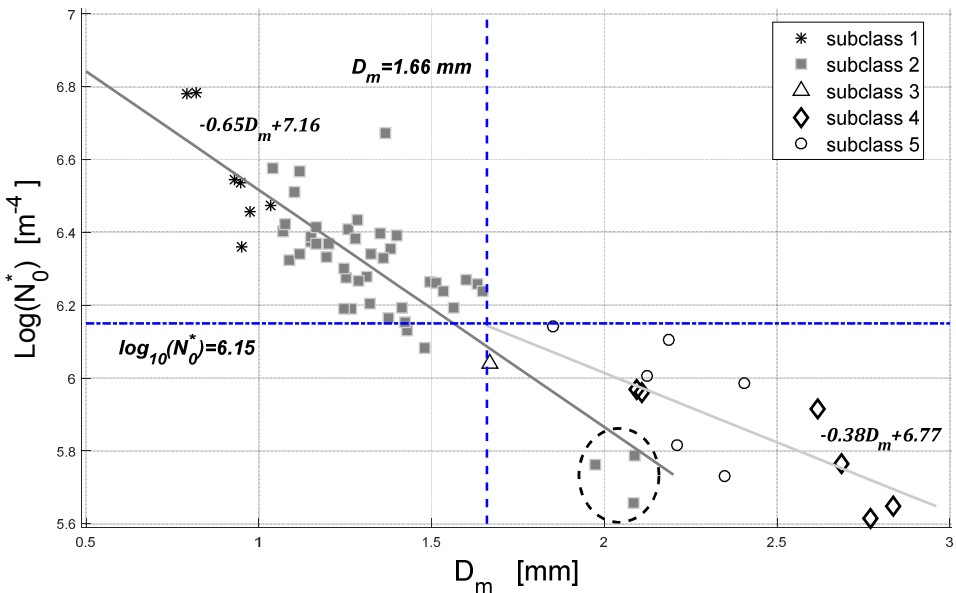

*Figure 9: Microphysical variable $N_0^*$ versus $D_m$ for the five rainy event subclasses. The three neurons corresponding to mixed events are circled. Dashed lines correspond to borders $D_m > 1.66$ and $Log(N_0^*) > 6.15$*