# Peer review of "Data-driven clustering of rain events: microphysics information derived from macro scale observations"

_Atmospheric Measurement Techniques, 2016_

## Referee Comment (RC1) · D. Dunkerley (Referee) · 29 Dec 2016

This paper presents an analysis of a set of disdrometer data (from 2008-2014) processed to yield a 1 minute rainfall series. This was then grouped into 545 simple rainfall events using an inter-event time of 30 minutes (page 3 line 31). Half of the events were used for 'learning' using some grouping methods, and the other half for testing the grouping methods.

From the literature, 17 characteristics of rainfall events (such a duration, depth, or mean rainfall rate) were explored as criteria for characterizing events. Since some of the characteristics are correlated, the authors sought to identify a smaller set of more parsimonious variables that might be sufficient to characterise rainfall events.

[Figure]

In terms of method and approach, the paper seems to be generally sound and to employ useful methods that are well applied. An aspect that I found to be missing was a discussion of the purpose for which rainfall events might need to be characterized. After all, it seems reasonable to think that this should guide the selection of parameters that might be meaningful for particular applications. For instance, in hydrology, it is well known that rainfall events cannot be characterized adequately through the use of simple descriptors such as average rainfall rate or peak intensity. Rather, measures of the time-distribution of rain and no-rain periods within an event are important, as is the position of the intensity peak(s) within the event – either early or late, for instance. These characteristics in turn affect the partitioning of water when rain arrives at the soil surface (e.g., far more of the rain tends to become overland flow if the largest intensity peak occurs late in the rainfall event, and much less becomes overland flow if the intensity peak is early).

A useful typology of 5 classes of event emerges from the numerical analyses presented in the paper, and I thought that this formed a significant and useful contribution of the paper. The link to convective and stratiform precipitation is well explored. However, here again, I felt that material was missing that might have been included in the paper. For instance, orographic rainfall can exhibit very different characteristics from other forms of precipitation, such as convective rain. This can include very long event durations and very prolonged rain of consistent or rising intensity. The authors give the impression that they only recognize convective and stratiform rainfall as end members. Likewise, they don't consider rainfall over the oceans, which has some temporal characteristics that distinguish it from terrestrial rainfall. As a result, I felt that the authors should offer some caveats about the extent to which they argue that their approach and conclusions might, or might not, be more widely applicable than to the single geographical location (and rainfall climate) represented by their disdrometer data. These caveats should be reflected in the title of the paper, which should be less sweeping, and perhaps refer to stratiform and convective rainfall, and/or to the particular study region (France) and its particular rainfall event characteristics. Of course, the authors

also neglect seasonal and inter-annual changes in rainfall event character, and this also warrants mention and possibly some discussion. The authors seem to consider that their results are in some way definitive and universally applicable. I doubt this, and would like them to consider how their results might have changed had they used a larger data set, including data from different rainfall climates.

I recommend some minor revision to address the above points.

The written English could be improved in places. For instance, 'criteria' is the plural form; when referring to a single parameter the word to use is 'criterion' (singular). 'Dysfunction' (page 3 line 36) should be 'malfunction'.

David Dunkerley Monash University

———————————————

---

## Referee Comment (RC2) · Anonymous Referee #2 · 5 Jan 2017

**1 Summary**

This manuscript proposes a data-driven approach to analyze rain-rate time series at the event time scale in order to link micro- and macro-physical properties of rainfall. After defining what is a rain event, a genetic algorithm is combined with a self-organizing map (SOM) method to identify the most informative descriptors of a rain-rate time series for a parsimonious approach. The obtained 5 descriptors are then used with the corresponding self-organizing map in order to "project" the initial 23-dimensional space into a 2-dimensional (map of neurons). An unsupervised clustering technique (hierarchical ascending clustering) is then applied to identify clusters in the neurons of the

[Figure]

SOM, corresponding to clusters of rainfall events. Using 2 clusters, the usual convective/stratiform dichotomy is retrieved, and the authors proposed to use up to 5 clusters. Taking advantage of the fact that the employed time series come from a disdrometer, the links between the rain descriptors or the neurons of the SOM and two important parameters of the drop size distribution (DSD) are investigated. In this way some relationships between rainfall macro and micro-physical properties are highlighted.

**2  Recommendation**

I enjoyed reading this manuscript (despite the quality of the English that must be improved) because the proposed approach is original and promising. Such characterization of rainfall events and the possible links between the macro and micro properties of rainfall are highly relevant to AMT readership and to the community in general. I have some relatively minor comments/suggestions listed below, I hence recommend to send the manuscript back tot the authors for minor revisions.

**3  General comments**

1. The dimensionality reduction is well explained, but I did not find a quantification of the amount of information lost in the process. The obtained 5 descriptors and the corresponding SOM are optimal with respect to the criterion defined (topological error), but this optimum could be bad in absolute term (i.e., a significant amount of information is lost overall even if the selected descriptors/SOM are better than other combinations of descriptors/SOMs). I missed such discussion in Sec.3.1.

2. How transferable to other climatic regions are the results obtained from the presented analyses? Can interested reader use the exact same SOM in other regions or it should be recomputed to adjust to the local climatology?

3. There are many grammar and vocabulary mistakes throughout the manuscript. The authors must have the manuscript edited by a professional or a native speaker at least. I cannot list all of them but here are a few examples: precipitation without s, clusterS (p.2, l.34), "In a second time" (p.2, l.36), punctual should be point (p.3, l.2), "is more able for detecting" (p.5, l.10), "the variables those the components" (p.5, l.36)...

**4  Specific comments**

1. Title: is microphysical information really derived from macrophysical information? Figure 9 shows that there is a link between a given neuron and $(N_w, D_m)$ but we do not know how the events "attached" to a given neuron are spread in the $(N_w, D_m)$ space.

2. P.1, l.38-39: dimensionality reduction implies more or less information loss. What can be discarded exactly may differ from one application to the other... Hence the intended application may be important.

3. P.2, l.8: microphysics does not reduce to the DSD (which corresponds more to the microstructure of rainfall). The use of microphysics in this context is a bit ambiguous and confusing.

4. P.2, l.13: there are more than a few disdrometers worldwide! Please rephrase.

5. P.2, l.20: some disdrometers allow the estimation of rain rate (and other variables) at higher temporal resolution than 1 min.

6. P.3, l.10: a rain event will also strongly depend on the considered spatial and temporal resolution. You work at the point scale, but a rain event could also

be defined over a given area (using model grids for instance). This should be mentioned I think.

7. P.3, l.30: how sensitive are the results to this MIT value of 30 min? As mentioned above, the spatial scale probably has an influence on the relevant MIT value.

8. P.4, l.7: the term "descriptors" could also be used here.

9. P.4, l.27: could you provide some quantitative information about the goodness-of-fit to the normal distribution of the different transformed variables? Are they close enough to Gaussian distribution?

10. P.4, l.33: "learning data set": it is not defined... I guess it is a subset of the total data sets, but how was it obtained?

11. P.4, l.33 - p.5, l.2: is this paragraph about PCA really necessary?

12. P.5, l.30: vector should be denoted in bold font.

13. P.5, l.32: why 60 chromosomes?

14. P.5, l.35: "training data set": same as above for learning set, it is not defined...

15. P.5, l.36-37: "Once training each ... variables": I do not understand this sentence, it seems there is a syntax issue.

16. P.6, l.6: could you provide the functional form of $te(x^k)$?

17. P.6, l.30: "describe quite well the original space": could you provide quantitative information on this aspect? Based on what can you state this?

18. P.6, l.28-32: if I am correct, the 5 selected variables are transformed ones, so their physical interpretation may be slightly less straightforward than suggested in the text.

19. P.7, l.21: why $8 \times 8$ neurons?

20. P.8, l.21-27: is this valid for all climatologies or just for the one studied here (temperate mid-latitudes)?

21. P.10, l.1: the delineation could be illustrated in Figure 6.

22. P.10, l.19: maybe you could add the coordinates (2,0.7) in the $(R_m, \beta_{L3})$ space to help the reader.

23. P.10, l.32: "body rain disturbances"?

24. P.12, l.11: given the definition provided in Eq.6, $D_m$ is the mass-weighted diameter.

25. The quality of figures 2, 3 and 6 should be improved.

---

## Referee Comment (RC3) · Anonymous Referee #3 · 11 Jan 2017

Paper revision. Title: Data driven clustering of rain events: microphysics information derived from macro scale observations. Author(s): M. D. Dilmi et al. MS No.: amt-2016-389

In this paper, the authors present a data-driven approach to analyze the rain events in order to characterize them both from a micro- and macro-physical point of view. The authors use a disdrometer dataset of 545 events (according to their definition of event), which is divided in two uneven groups, one for learning and on for testing. A Genetic Algorithm (GA) is combined with a self-organizing map (SOM) method to identify a number of indicators (from a list of 23 micro- and macro-variables) able to full describe each event. A previous step led to identify the independent indicators though a

[Figure]

Principal Component Analysis (PCA). They found that the best performance is obtained by using 5 macro indicators. A hierarchical cluster analysis is then applied to the 5 indicators subset to cluster all the events in the two "commons" convective/stratiform groups and in five finer groups.

I found the manuscript interesting because the authors use a new technique (moreover derived from a different science field) and even because they aimed to link micro- and macro-physical variables to describe a rain event. Generally, the quality of figures and tables is good, but I suggest to box all the figures and to increase the quality of figure 1a. However, some questions and doubts arise reading the manuscript. I suggest the publication of the manuscript after the authors address the following points.

• Generally, the English has to be improved. Several errors are found by reading the text and I suggest you to review the manuscript by a native speaker. Some errors are very basic and could be just writing error (i.e. "is justify" Pag.1 line 16, the lacking of "s" in some third person verbs, etc.). Please revise the expression like "one can see/consider/note. . .." and check the correct use of the singular/plural words.

• What I mainly miss within the manuscript are two aspects: what does it happen if the Minimum Inter-event Time (MIT) is chosen different from 30 minutes? What I ask here is if the authors have been conducted a sensitivity study to show the influence of the MIT on the results. This could be very useful if the technique is applied to a different definition of event. The second point is related to the measurement instrument. All their analysis are based on disdrometer data that are not so widespread with respect to the rain gauges. Moreover, they found that the best performance is obtained by five macrophysical indicators, which are also a rain gauge outputs. So, do the authors think that they get the same results if the rain gauges data are analyzed? It could be interesting, if they have some rain gauges data available, to apply their technique to this type of data. It is well known that rain gauges have some problems in measuring very light and heavy precipitation. How this can impact on the results?

• In the section 3.1, the authors describe the methodology used. Even if it is quite well explained, I suggest to slightly improve so that every reader can be able to correctly reproduce it (i.e. they should better explain what is the topological error). Minor comments: • Page 2, line 13: "Around the world, there are few disdrometers. . .". The expressions is incorrect even if I understand the will of the authors to highlight the high ratio between the number of disdrometers.

• Figure 3: I do not understand the colorbar values. Can you better explain them?

[Figure]

---

## Referee Comment (RC4) · Anonymous Referee #4 · 15 Jan 2017

Brief Summary/General Comments:

This manuscript develops an approach to analyze and interpret rain disdrometer data by using statistical and analytical tools common in other scientific communities, but relatively unknown within the hydrological and rain microphysical communities.

As a frequent user of disdrometric data, I find the analysis presented in this paper to be fascinating and potentially transformative to future work in hydrology and rain microphysics. It is clear the methods presented here are extremely novel within this community, and it appears that there is a lot of valuable information that can possibly be gained by conducting analyses in the ways described within the text.

[Figure]

However, as other referees have noted, the entire manuscript needs a thorough re-vision by native English speakers in the interest of readability. Since the analytical methods used here are expected to be new to most of the intended readership, it is vitally important that the exposition is clear, unambiguous, and complete enough so that others can follow in the footsteps of these investigators; the tone of these sections would be most useful to the reader if they were very tutorial in nature. In particular, sections 3 and 4 (which involved fundamentally new ideas for me) require more detail and clarity to allow a reader to reproduce and/or mimic the work with similar data-sets.

As it stands, I only have a vague understanding of the details of the process that was undertaken – but the results and methods are intriguing enough that I desperately wish the manuscript could be rewritten in such a way that a diligent reader could reproduce the results. The text is currently opaque enough that it is difficult for me to determine whether or not there are technical issues related to the work.

Specific Comments:

In addition to the presentation concerns outlined above (and by the other reviewers), I do have some basic scientific questions to add to the discussion.

1. As noted by other reviewers, I am curious as to how much the choice of a 30-minute MIT interval used to define a rain event effects the results. Similarly, how much does the minimum detection thresh-hold influence the results?

2. Given that this method is new to some readers, could it briefly be described what happens with this method if it is employed with non-transformed data? The normaliza-tion method used to make each of the variables pseudo-normal sounds quite practical, but I always wonder what biases this can introduce in fundamental non-linear pro-cesses like rainfall. What would happen if such a transformation is not utilized?

3. The instrument you utilized had a 1-minute integration time. There are other de-tectors with sub-second integration times, and one can always coarsen data. What influence does integration time have on the method proposed? Does one find the same basic results (for the entire process if the full analysis is done with a different integration time-scale? This is not merely of academic interest here, since scale-matching is of vital importance within the hydrological community where instruments with very coarse resolution (e.g. radar) are often "ground-truthed" with point-detector measurements that can have sub-millisecond resolutions.

4. As noted by other reviewers, the conclusions seem a bit of an over-reach for a study done using one type of instrument, with one type of MIT choice, one minimum detection threshhold, at one location, with one integration time. I believe that if this approach was applied by a number of investigators at different locations using data from a number of different instruments and a variety of parameter choices it is possible that a very powerful result could be attained. I would love to work on such a study, but – due to the quality of written English in this paper – I can't follow the method closely enough to do so at this time.

I would like to petition the authors to give this paper a substantial re-write and have native English speakers edit it. In the interest of readability, I would encourage figure captions to be more descriptive. Finally, I would love to have access to either code or other materials necessary to replicate the work (so, as other reviewers have pointed out, unspecified details like the choices made in te($x\hat{}k$) are known to a practitioner).

If these changes could be implemented, I would promise the authors that not only will this paper be read, but some of us will use the results in their own work – myself included.

---

## Referee Comment (RC5) · A. Parodi (Referee) · 15 Jan 2017

The authors present an interesting study on data driven clustering of rain events based on the analysis of characteristics related to rain rates or rain accumulation (macro physical information) and to the raindrop size distribution (microphysical information).

The research results are interesting, relevant and timely.

Some questions, and possibly suggestions for future work, are however presented hereafter:

1) the authors adopt a Minimum Inter-event Time (MIT) of 30 minutes. Did they perform any sensitivity of the presented results versus lower (15 minutes) and higher (60 and

120 minutes) threshold?

2) a total of 23 indicators for macro physical description of rain events were defined, however the references provided in order to support their choices are rather limited. Please improve.

3) also the details provided about the 23 indicators computation are rather poor. Please improve

4) by checking the position of the test site, namely "Site Instrumental de Recherche par Télédétection Atmosphérique" (SIRTA1) in Palaiseau (France), i noticed that it is not far from the Trappes sounding site. Then i was wondering if the authors plan to explore possible relationship between the identified characteristics related to rain rates or rain accumulation (macro physical information) and to the raindrop size distribution (microphysical information) versus the vertical thermodynamical structure observed for the identified events in the period 2008-2014.

For example

Molini, L., Parodi, A., Rebora, N., Craig, G. C. (2011). Classifying severe rainfall events over Italy by hydrometeorological and dynamical criteria. Quarterly Journal of the Royal Meteorological Society, 137(654), 148-154.

For about 81 events, a time-scale for convective adjustment was computed, based on gridded hourly precipitation rates derived from rain-gauge data and ECMWF analysis (ERA-Interim) of convective available potential energy (CAPE). Values of the convective adjustment time-scale, $\tau c$, shorter than 6 h indicate convection that is responding rapidly to to the synoptic environment (equilibrium), while slower time-scales indicate that other, presumably local, factors dominate.

It would be interesting to see if and how a local convective adjustment time-scale (computed using Trappes data) is related to the results of this study.

5) in a recent paper

Bühl, J., Leinweber, R., Görsdorf, U., Radenz, M., Ansmann, A., Lehmann, V. (2015). Combined vertical-velocity observations with Doppler lidar, cloud radar and wind profiler. Atmospheric Measurement Techniques, 8(8), 3527-3536.

it is explored the potential of combined vertical-velocity observations with Doppler lidar, cloud radar and wind profiler.

In this respect, even if the sodar located at the Charles de Gaulle Airport is relatively far away (50 km), did the authors consider as possible to explore relationship (if any) between updraft and downdraft velocity versus the microphysical analysis performed in this study?

I would be very curious to test, in a real-word situation, the results we got years ago in these papers about the relationship between raindrop diameter and updraft velocities:

Parodi, A., Emanuel, K. (2009). A theory for buoyancy and velocity scales in deep moist convection. Journal of the Atmospheric Sciences, 66(11), 3449-3463.

Parodi, A., Foufoula Georgiou, E., Emanuel, K. (2011). Signature of microphysics on spatial rainfall statistics. Journal of Geophysical Research: Atmospheres, 116(D14).

---

## Author Comment (AC1) · 6 Feb 2017

D. Dunkerley (Referee)
david.dunkerley@monash.edu

- 	This paper presents an analysis of a set of disdrometer data (from 2008-2014) processed to yield a 1 minute rainfall series. This was then grouped into 545 simple rainfall events using an inter-event time of 30 minutes (page 3 line 31). Half of the events were used for 'learning' using some grouping methods, and the other half for testing the grouping methods.

From the literature, 17 characteristics of rainfall events (such a duration, depth, or mean rainfall rate) were explored as criteria for characterizing events. Since some of the characteristics are correlated, the authors sought to identify a smaller set of more parsimonious variables that might be sufficient to characterise rainfall events.

In terms of method and approach, the paper seems to be generally sound and to employ useful methods that are well applied.

An aspect that I found to be missing was a discussion of the purpose for which rainfall events might need to be characterized. After all, it seems reasonable to think that this should guide the selection of parameters that might be meaningful for particular applications. For instance, in hydrology, it is well known that rainfall events cannot be characterized adequately through the use of simple descriptors such as average rainfall rate or peak intensity. Rather, measures of the time-distribution of rain and no-rain periods within an event are important, as is the position of the intensity peak(s) within the event – either early or late, for instance.

These characteristics in turn affect the partitioning of water when rain arrives at the soil surface (e.g., far more of the rain tends to become overland flow if the largest intensity peak occurs late in the rainfall event, and much less becomes overland flow if the intensity peak is early).

*We agree on the fact that, given an application, some parameters clearly carry more information than others. This article mainly aims at validating a methodology able to select the most parsimonious subset of variables representative of the whole. The 23 parameters of this study are clearly not exhaustive. They encompass the parameters usually used by the authors themselves, to which are added those derived from a bibliographic study. They nevertheless allow to validate the methodology. In further studies it will be possible to improve the set of parameters to test their ability to better characterize the rain events. It is also possible to adapt the parameter set with a given application in mind.*

*Often parameters are the result of the knowledge of various given physical/environmental contexts. When dealing with a data set, the researcher often has to relate his case to the numerous situations encountered in the literature. It can be puzzling to choose the most appropriate parameters. The methodology allows to automatically bring out a reduced set of pertinent parameters. This article illustrates a way to discriminate among numerous parameters.*

*The SOM algorithm implies that each neuron of the map gather similar data, rain events in our case. By selecting the best map according to the topological error we make sure that the events gathered by a neuron are close to the events of its neighbors.  By doing so we make sure that the map is well spread in the data space.*
*The map can be seen as a classification of the rain events. The five parameters we hold on to are the one best minimal subspace preserving the topology and thus the similarity in the original data space.*
*The underlying hypothesis is that the most valuable information is the one which best represent the similarity/dissimarity in the original data space.*
*This assumption is based on the fact that the parameters are redundant and sufficiently coherent to learn a meaningful map. This is validated by the validation process.*

**The following sentence is added p.4 L7**
**Of course this set of 17 features is not exhaustive and some other features could be added depending the application. For example is hydrology the positions of the intensity peaks inside the event could be a relevant feature.**

**The following sentence p1 L36  is modified**

The first goal of this study is to select the most relevant features to characterize the events through a data-driven approach without taking into account the application context and thus to characterize a rain event in the most parsimonious and efficient possible way.

**New formulation**

*The first goal of this study is to select the most relevant features to characterize the events through a data-driven approach without taking into account the a priori knowledges of the field of application and thus to characterize a rain event in the most parsimonious and efficient possible way.*

-       A useful typology of 5 classes of event emerges from the numerical analyses presented in the paper, and I thought that this formed a significant and useful contribution of the paper. The link to convective and stratiform precipitation is well explored. However, here again, I felt that material was missing that might have been included in the paper. For instance, orographic rainfall can exhibit very different characteristics from other forms of precipitation, such as convective rain. This can include very long event durations and very prolonged rain of consistent or rising intensity. The authors give the impression that they only recognize convective and stratiform rainfall as end members.

Likewise, they don't consider rainfall over the oceans, which has some temporal characteristics that distinguish it from terrestrial rainfall. As a result, I felt that the authors should offer some caveats about the extent to which they argue that their approach and conclusions might, or might not, be more widely applicable than to the single geographical location (and rainfall climate) represented by their disdrometer data. These caveats should be reflected in the title of the paper, which should be less sweeping, and perhaps refer to stratiform and convective rainfall, and/or to the particular study region (France) and its particular rainfall event characteristics. Of course, the authors also neglect seasonal and inter-annual changes in rainfall event character, and this also warrants mention and possibly some discussion. The authors seem to consider that their results are in some way definitive and universally applicable. I doubt this, and would like them to consider how their results might have changed had they used a larger data set, including data from different rainfall climates.

*We agree, we were not clear enough on the purpose of this work. The conclusions drawn in this article follow from the data set used in the article which is representative of a region (Ile de France in France). There is no* orographic rainfall *events*

*Given the size of the data set we could not take seasonal and inter-annual changes in rainfall event.*

*When parting  the complete rain event set in two. To insure a correct generalization we made sure to take different time periods. When looking closely to the data, we notice that the 2 periods are noticeably different which ensure a better generalization.  With a larger data set, it would be interesting to study the seasonal and the inter-annual information encompassed in the map.*

*Of course novel data such as orographic and oceanic rain events would probability not be as well taken into account by the map. Nevertheless, the map was thoroughly validated. By making sure that the topology was preserved we made sure that the behavior of the map would be as good as possible.*

*Even if it is far from exhaustive the data set gather a large variety of events. If the map was learned on a more exhaustive data set first the number of data would increase implying the possible use of a bigger map. It would also imply a richer panel of data most certainly with a wealth of behavior not present in the current data set. In this case, new parameters better characterizing these new rain events ought to be added to highlight their specific properties.*

*For example, concerning a very long and consistent Orographic rain event,  the current five parameters stressed out in the article may be enough to characterize such an event. If we were to take into account something as specific as a rising intensity a dedicated parameter ought to be added.*

*Adding new data should modify the data density allowing the rise of a neuron specific of a new class of events. Stressing the preservation of the topology in the algorithm is done in this sense.*

*Once the methodology validated and vetted, we are going to work on larger data sets potentially exhaustive. It is a work in itself.*

*The following sentences were added in the conclusion : see below*

**Some caveats were added**
**p11 L36 : …. But sufficiently heterogeneous between them. Of course this classification is obtained for middle latitude climate. The data set used in this study is only representative of a particular region and topography (Ile de France in France, temperate climate). Data driven analysis cannot lead information on process that are not sampled in the data set, there are no orographic rainfall events nor oceanic observations which could present particular features and lead to additional specific clusters of events. The last step …..**

**P15 L18 (conclusion) : The present study was conducted with observations from middle latitude area in plain region. The relevance of this classification needs to be confirmed with other data set collected in different climatic areas and for different meteorological situations encountered for example in mountain or coastal areas. If the SOM was learned on a more exhaustive data set the number of data would increase allowing the use of a bigger map allowing to represent possible new behaviors which are not present in the current data set. This point will have to be addressed in future works**

- I recommend some minor revision to address the above points.

The written English could be improved in places. For instance, 'criteria' is the plural form; when referring to a single parameter the word to use is 'criterion' (singular).
'Dysfunction' (page 3 line 36) should be 'malfunction'.

*We agree. A thorough revision of the manuscript will be done by a professional translator.*

---

## Author Comment (AC2) · 6 Feb 2017

1 Summary

This manuscript proposes a data-driven approach to analyze rain-rate time series at the event time scale in order to link micro- and macro-physical properties of rainfall. After defining what is a rain event, a genetic algorithm is combined with a self-organizing map (SOM) method to identify the most informative descriptors of a rain-rate time series for a parsimonious approach. The obtained 5 descriptors are then used with the corresponding self-organizing map in order to "project" the initial 23-dimensional space into a 2-dimensional (map of neurons). An unsupervised clustering technique (hierarchical ascending clustering) is then applied to identify clusters in the neurons of the SOM, corresponding to clusters of rainfall events. Using 2 clusters, the usual convective/stratiform dichotomy is retrieved, and the authors proposed to use up to 5 clusters.
Taking advantage of the fact that the employed time series come from a disdrometer, the links between the rain descriptors or the neurons of the SOM and two important parameters of the drop size distribution (DSD) are investigated. In this way some relationships between rainfall macro and micro-physical properties are highlighted.

2 Recommendation

I enjoyed reading this manuscript (despite the quality of the English that must be improved) because the proposed approach is original and promising. Such characterization of rainfall events and the possible links between the macro and micro properties of rainfall are highly relevant to AMT readership and to the community in general. I have some relatively minor comments/suggestions listed below, I hence recommend to send the manuscript back to the authors for minor revisions.

3 General comments

1. The dimensionality reduction is well explained, but I did not find a quantification of the amount of information lost in the process. The obtained 5 descriptors and the corresponding SOM are optimal with respect to the criterion defined (topological error), but this optimum could be bad in absolute term (i.e., a significant amount of information is lost overall even if the selected descriptors/SOM are better than other combinations of descriptors/SOMs). I missed such discussion in Sec.3.1.

*The SOM algorithm aims at giving a low quantization error. First the map, is learned with a coarse training allowing the map to spread well (with no folding) in the data space, keeping the topology of the original data space. The second step, fine tuning, insures that the neurons become local averages of the data. This step insures a good quantization of the data.*
*The final map is a compromise between the topological error and the number of parameters. Since the topological error is computed on the total data space, it insures that the parameters used to learn the map also preserved the topology for most of the other parameters. This insures that data close with respect to the five final parameters are also close in the full/original data space. For parameters used to characterize natural events, like rain, ruled by the law of physics this implies that the map was able to learn the relationship between the parameters (even for the parameters not used in the mapping of the data space), otherwise the topology would not be preserved.*

*Unlike linear approaches, like principal component analysis, that uses an orthogonal transformation, the non-linear dimensionality reduction method uses a mapping quantification of the amount of information. A quantification of the amount of information lost in the process is quite complex to evaluate due to nonlinear relationships between variables. For this reason we have provided in table 4 the R2 determination coefficient for the whole variables. As it can be seen in this table the determination coefficient are rather good even for the unlearned variables. This guarantees the good behavior of the map and consequently only few information is lost.*

2. How transferable to other climatic regions are the results obtained from the presented analyses? Can interested reader use the exact same SOM in other regions or it should be recomputed to adjust to the local climatology?

*Of course novel data such as orographic and oceanic rain events would probability not be as well taken into account by the map. Nevertheless, the map was thoroughly validated. By making sure that the topology was preserved we made sure that the behavior of the map would be as good as possible. This point will have to be addressed in future work*

**The following sentences were added :**
**p11 L36 : …. But sufficiently heterogeneous between them. Of course this classification is obtained for mid latitude climate. The data set used in study is only representative of a particular region and topography (Ile de**

**France in France, mid latitude climate). Data driven analysis can not lead information on process that are not sampled in the data set, there are no orographic** rainfall **events nor oceanic observations which could present particular features and lead to additional specific clusters of events.. The last step …..**

**P15 L18 (conclusion) : The present study was conducted with observations from middle latitude area in plain region. The relevance of this classification needs to be confirmed with other data set collected in different climatic areas and for different meteorological situations encountered for example in mountain or coastal areas. This point will have to be addressed in future work.**

3. There are many grammar and vocabulary mistakes throughout the manuscript.
The authors must have the manuscript edited by a professional or a native speaker at least. I cannot list all of them but here are a few examples: precipitation without s, clusterS (p.2, l.34), "In a second time" (p.2, l.36), punctual should be point (p.3, l.2), "is more able for detecting" (p.5, l.10), "the variables those the components" (p.5, l.36)...

*We agree. A professional translator will edit the manuscript.*

4 Specific comments

1. Title: is microphysical information really derived from macrophysical information?
*It is mean that some microphysical properties can be inferred thank to the knowledge of the macrophysics subclass. Hence an event belonging to subclass 4 has high Dm (2.7 mm) and low N0\* (5.7)*

Figure 9 shows that there is a link between a given neuron and (Nw , Dm ) but we do not know how the events "attached" to a given neuron are spread in the (Nw , Dm ) space.

*Here is Figure 9 with added data points. The points are colored with the color of the class they belongs to. We can see that for each class the corresponding neurons are quite well spread among the data belonging to the same class.*

[Figure]

2. P.1, l.38-39: dimensionality reduction implies more or less information loss. What can be discarded exactly may differ from one application to the other... Hence the intended application may be important.

*The dimension reduction indeed implies potential information loss. Nonetheless, the large number of parameters implies redundancy. Dimensionality reduction is intended to reduce this redundancy thus part of the information loss can be considered as denoising.*

*P1, l.38, we replaced "to select the most relevant features" by "to select a reduced set of features"*

*Concerning what can be discarded or not, let's consider the unlearned IETp variable because we knew that it would not bring pertinent information on a rain event.. The resulting map is not structured with respect to the IET which is, as it should be expected, considered as noise. Thus any loss of information associated to an impertinent parameter cannot be considered in the end as a loss of information. An impertinent parameter even if learned is inconsistent with the rest of the information used to describe an event. It is thus discarded during the learning process. Hence the resulting map is not structured according to this variable.*

*We intended a general approach aimed toward the retrieval of microphysical information. We believe that this approach is generic and that the resulting map, due to the specificity of the algorithm, should also be useful for other applications. Either way, for a specific application, if a dedicated data set is provided, it should be used to apply the algorithm.*

3. P.2, l.8: microphysics does not reduce to the DSD (which corresponds more to the microstructure of rainfall). The use of microphysics in this context is a bit ambiguous and confusing.

We agree. These sentences :
"**Usually the microphysics is characterized by** the raindrop size distribution, noted N(D), which is defined by the number of raindrops per unit of volume and per unit of raindrop diameter (D). Actually, information on **rain microphysics** is often displayed through proxies **of N(D)** as it will be explained further. At the present time, **the rain microphysics** features are not accessible by rain gauges which only provide macrophysical information."
*are replaced by:*
"**One of the key information for remote sensing is** the raindrop size distribution, noted N(D), which is defined by the number of raindrops per unit of volume and per unit of raindrop diameter (D). Actually, information on **raindrop size distribution** is often displayed through **its** proxies as it will be explained further. At the present time, **these kinds of** features are not accessible by rain gauges which only provide macrophysical information."

4. P.2, l.13: there are more than a few disdrometers worldwide! Please rephrase.

*We agree.*
"**Around the world, there are few disdrometers where there are tens of thousands of rain gauges.**"
*is replaced by*
"**Around the world, there are tens of thousands of rain gauges where there are a lot less disdrometers.**"

5. P.2, l.20: some disdrometers allow the estimation of rain rate (and other variables) at higher temporal resolution than 1 min.

"**Disdrometers provide drop size distribution and consequently they allow estimating one-minute rain rates which are the measurements used to get hydrological information.**"

*is replaced by:*

"**Disdrometers provide drop size distribution and consequently they allow estimating one-minute (or less) rain rates which are the measurements used in this study to get hydrological information.**"

6. P.3, l.10: a rain event will also strongly depend on the considered spatial and temporal resolution. You work at the point scale, but a rain event could also be defined over a given area (using model grids for instance). This should be mentioned I think.

*"Indeed a rain event will depend on the sensor characteristics (specific surface caption, detection threshold, instrumental noise)."*
*is replaced by:*
"**Indeed a rain event will depend on the sensor characteristics (specific surface caption, detection threshold, instrumental noise) as well as on the spatial or temporal resolution chosen for the study.**"

7. P.3, l.30: how sensitive are the results to this MIT value of 30 min? As mentioned above, the spatial scale probably has an influence on the relevant MIT value.

*We have run some quick test for various values of the MIT, but we have not really investigated this point and consequently we are not really in position to answer to this question. A greater MIT value will aggregate rain periods together and consequently will modify some parameters like the event duration 'De' or the Rain event Depth 'Rd'. Some others parameters can remain unchanged or not, this is the case for example for parameters Pci (which is sensitive to the variability of the rain rate). We expect that a higher MIT may aggregate events of a different type, whereas a shorter MIT tends to increase the number of events while they belong to the same group of rain cells*

8. P.4, l.7: the term "descriptors" could also be used here.

*Changed*

9. P.4, l.27: could you provide some quantitative information about the goodness-of-fit to the normal distribution of the different transformed variables? Are they close enough to Gaussian distribution?

*The transformations are empirical. Our algorithm or the PCA do not rely on the gaussianity of the data. Properly speaking, there is no statistical or probabilistic framework in this article.*
*(No estimator, no confidence interval or no whatsoever are involved.)*
*As it is written P.4, l.25 and P.4, l.27 the final distributions are "quasi-normal" or "the closest to a normal distribution". As it is stated P.4, l.26 the transformations were chosen "empirically" on the basis of the kurtosis and skewness estimation. Before using any learning algorithm, the data must be checked and transformed to be fitted for the algorithm. Here the transformations are simply used to contract/dilate the data space to guarantee a better spreading of the map among the data.*
*Having transformed distributions, close to normal, also helps for the interpretation of the PCA.*

10. P.4, l.33: "learning data set": it is not defined... I guess it is a subset of the total data sets, but how was it obtained?

*The learning and test sets are defined at the end of section 2.1 where a reference is made to the Table Tab.1..*

*"on the learning data set."*
*is replaced by*
*"on the learning data set (cf. end of section 2.1 and Tab.1)."*

11. P.4, l.33 - p.5, l.2: is this paragraph about PCA really necessary?

*The PCA is a well-established technique known from most. Many readers may wonder what could have been done with a PCA. Since it could be counterproductive the PCA is done. It is also a way to display the redundancy in the parameters. It also allows to introduce the fact that some parameters (such as IETp) are potentially non-informative.*

12. P.5, l.30: vector should be denoted in bold font.
ok

13. P.5, l.32: why 60 chromosomes?
*This value is not critical and any even value can be used. A smaller number of chromosomes involves more generations to converge. The GA theory shows that the algorithm converges to the same solution whatever the number of chromosomes.*

14. P.5, l.35: "training data set": same as above for learning set, it is not defined...
Training data set is replaced by learning data set

15. P.5, l.36-37: "Once training each ... variables": I do not understand this sentence, it seems there is a syntax issue.

*There is indeed a syntax issue.*
**"Once training each of the 60 Maps"**
*is replaced by:*
**"Once trained each of the 60 Maps"**

16. P.6, l.6: could you provide the functional form of te(xk)?
*Additional information is now provided in sections 3 and 4.*
*the topological error $te(x^k)$ is computed according to eq. 2 in Uriarte and Martín (2008)*

*E. Arsuaga Uriarte, and F. Díaz Martín, Topology Preservation in SOM, World Academy of Science, Engineering and Technology. International Journal of Computer, Electrical, Automation, Control and Information Engineering Vol:2, No:9, 2008*

17. P.6, l.30: "describe quite well the original space": could you provide quantitative information on this aspect? Based on what can you state this?

*The discussion concerning the representativeness of the 5 selected variables is discussed in sections 4.1 & 4.2. Consequently this sentence has nothing to do here. The sentence is replaced by the following one :*

**At the 187th generation we get a subspace formed by 5 variables namely : Event duration $D\_e$ (#1), Standard deviation $\sigma\_R$ (#9), rain rate peak in event $R\_{max}$ (#11), Rain event depth $R\_d$ (#13), Absolute rain rate variation $P\_{c1}$ (#14).**

18. P.6, l.28-32: if I am correct, the 5 selected variables are transformed ones, so their physical interpretation may be slightly less straightforward than suggested in the text.

*As told previously the transformations are simply used to contract/dilate the data space to guarantee a better spreading of the map among the data. It has an impact on the distances between the data. It does not affect the ordering of the data. It will not impact the interpretation of the map.*

19. P.7, l.21: why 8 × 8 neurons?

*64 neurons is a compromise. When we use the topological error we aim at a fine mapping of the data space. A smaller map would imply a poorer description of the data space. We have 234 rain events. 64 neurons imply more or less 3 to 4 data points to compute a neuron referent. We could not have more neurons. Choosing a square map implied not making hypotheses on the shape of the map that could have been representative only of our non-exhaustive data set.*

20. P.8, l.21-27: is this valid for all climatologies or just for the one studied here (temperate mid-latitudes)?

*It is not valid for all climatologies. The text has been clarified.*
*"These authors have noticed that successive rain and no rain periods are found to be uncorrelated, thus a rain time series can be considered by an alternation of rain event and no rain independently drawn periods. That is similar to say that inter-event time (IET) doesn't characterize the rain events. Brown et al. (1983) also investigated a possible dependence between IETp and the intra-event characteristics and they conclude that the assessment of their data gave no indication that such dependency exists."*
*is replaced by:*
**"These authors, working in temperate mid-latitudes for relatively short periods, have noticed that successive rain and no rain periods are found to be uncorrelated, thus a rain time series can be considered by an alternation of rain event and no rain independently drawn periods. That is similar to say that inter-event time (IET) doesn't characterize the rain events. For other places and other climatologies it is less clear, Brown et al. (1983) also investigated a possible dependence between IETp and the intra-event characteristics and they conclude that the assessment of their data gave no indication that such dependency exists."**

21. P.10, l.1: the delineation could be illustrated in Figure 6.

*Added*

22. P.10, l.19: maybe you could add the coordinates (2,0.7) in the ($R_m$, $\beta L3$ ) space to help the reader.

*It has been modified.*

23. P.10, l.32: "body rain disturbances"?

*It has been corrected.*
"the synoptic precipitations caused by the mid-latitude depressions are an example of stratiform precipitations. *They manifest themselves in the body rainy disturbances associated with warm and cold fronts."*
*is replaced by:*

"the synoptic precipitations caused by the mid-latitude depressions are an example of stratiform precipitations **they manifest themselves in disturbances due to warm and cold fronts.**"

24. P.12, l.11: given the definition provided in Eq.6, Dm is the mass-weighted diameter.

*You are right.*
*"This variability is represented by two microphysics parameters namely the* mean volume *diameter (Dm) and the parameter N\*."*
*is replaced by:*
**"This variability is represented by two microphysics parameters namely the** mass-weighted **diameter (Dm) and the parameter N\*."**

25. The quality of figures 2, 3 and 6 should be improved.

*We agree. The resolution has been improved and the size of the font changed.*

---

## Author Comment (AC3) · 6 Feb 2017

Paper revision. Title: Data driven clustering of rain events: microphysics information derived from macro scale observations. Author(s): M. D. Dilmi et al. MS No.: amt-2016-389

In this paper, the authors present a data-driven approach to analyze the rain events in order to characterize them both from a micro- and macro-physical point of view.
The authors use a disdrometer dataset of 545 events (according to their definition of event), which is divided in two uneven groups, one for learning and on for testing.
A Genetic Algorithm (GA) is combined with a self-organizing map (SOM) method to identify a number of indicators (from a list of 23 micro- and macro-variables) able to full describe each event. A previous step led to identify the independent indicators though a Principal Component Analysis (PCA). They found that the best performance is obtained by using 5 macro indicators. A hierarchical cluster analysis is then applied to the 5 indicators subset to cluster all the events in the two "commons" convective/stratiform groups and in five finer groups.

I found the manuscript interesting because the authors use a new technique (moreover derived from a different science field) and even because they aimed to link micro- and macro-physical variables to describe a rain event. Generally, the quality of figures and tables is good, but I suggest to box all the figures and to increase the quality of figure 1a. However, some questions and doubts arise reading the manuscript. I suggest the publication of the manuscript after the authors address the following points.

*We partially agree. The figure 1 was re-drawn with a better resolution and boxed. We chose not to box the other figures.*

 - Generally, the English has to be improved. Several errors are found by reading the text and I suggest you to review the manuscript by a native speaker. Some errors are very basic and could be just writing error (i.e. "is justify" Pag.1 line 16, the lacking of "s" in some third person verbs, etc.). Please revise the expression like "one can see/consider/note. . .." and check the correct use of the singular/plural words.

*We agree. A thorough revision of the manuscript will be done by a professional translator.*
.

- What I mainly miss within the manuscript are two aspects: what does it happen if the Minimum Inter-event Time (MIT) is chosen different from 30 minutes? What I ask here is if the authors have been conducted a sensitivity study to show the influence of the MIT on the results. This could be very useful if the technique is applied to a different definition of event.

*Our choice was made based on the literature. Even if it cannot be called a sensitivity study,  we also run some test to see if the conclusion drawn in the lite nevertheless literature were coherent with our data set.*
*We have not really investigated this point and consequently we are not really in position to answer to this question. A greater MIT value will aggregate rain periods together and consequently will modify some parameters like the event duration De or the Rain event Depth Rd. Some others parameters can remain unchanged or not, this is the case for example for parameters Pci (which is sensitive to the variability of the rain rate). We expect that a higher MIT may aggregate events of a different type, whereas a shorter MIT tends to increase the number of events while they belong to the same group of rain cells*

The second point is related to the measurement instrument. All their analysis are based on disdrometer data that are not so widespread with respect to the rain gauges. Moreover, they found that the best performance is obtained by five macrophysical indicators, which are also a rain gauge outputs. So, do the authors think that they get the same results if the rain gauges data are analyzed? It could be interesting, if they have some rain gauges data available, to apply their technique to this type of data. It is well known that rain gauges have some problems in measuring very light and heavy precipitation. How this can impact on the results?

*It is one of the perspectives of this work. It would give access to more exhaustive (as well in space and in time) data sets. We are working on this aspect. It has to be noticed that the nature of the measurements is different. (For example, a tipping bucket rain gauge is measuring the time to fill a given volume. It has some important implications for stratiform rain events.)*
*Since the resolution is less good it will definitely impact the results. It will strongly impact some parameters such as the Rmax and the PC parameters. Since Rmax and PC$_1$ are two of the five parameters selected by the algorithm it will impact the results. This is nevertheless a study work in itself.*

- In the section 3.1, the authors describe the methodology used. Even if it is quite well explained, I suggest to slightly improve so that every reader can be able to correctly reproduce it (i.e. they should better explain what is the topological error).

*We agree. Section 3 has been partially re-written.*

Minor comments:

- Page 2, line 13: "Around the world, there are few disdrometers. . .".
The expressions is incorrect even if I understand the will of the authors to highlight the high ratio between the number of disdrometers.

**The sentence :**
**"Around the world, there are few disdrometers where there are tens of thousands of rain gauges."**
*is replaced by*
**"Around the world, there are tens of thousands of rain gauges where there are a lot less disdrometers."**

- Figure 3: I do not understand the colorbar values. Can you better explain them?
*The colorbar represents the average distance between a particular neuron and its neighbors*
*The caption of Fig. 3 is modified:*
**The color of each neuron represents the average distance between the neuron and its neighbors.**

---

## Author Comment (AC4) · 6 Feb 2017

Brief Summary/General Comments:

This manuscript develops an approach to analyze and interpret rain disdrometer data by using statistical and analytical tools common in other scientific communities, but relatively unknown within the hydrological and rain microphysical communities.

As a frequent user of disdrometric data, I find the analysis presented in this paper to be fascinating and potentially transformative to future work in hydrology and rain microphysics. It is clear the methods presented here are extremely novel within this community, and it appears that there is a lot of valuable information that can possibly be gained by conducting analyses in the ways described within the text.

However, as other referees have noted, the entire manuscript needs a thorough revision by native English speakers in the interest of readability. Since the analytical methods used here are expected to be new to most of the intended readership, it is vitally important that the exposition is clear, unambiguous, and complete enough so that others can follow in the footsteps of these investigators; the tone of these sections would be most useful to the reader if they were very tutorial in nature. In particular, sections 3 and 4 (which involved fundamentally new ideas for me) require more detail and clarity to allow a reader to reproduce and/or mimic the work with similar data-sets.

As it stands, I only have a vague understanding of the details of the process that was undertaken – but the results and methods are intriguing enough that I desperately wish the manuscript could be rewritten in such a way that a diligent reader could reproduce the results. The text is currently opaque enough that it is difficult for me to determine whether or not there are technical issues related to the work.

**Responses to general comments**

*In section 3, we have added more detailed information and references concerning the variables selection. The figure 2 has been enlarged and made more readable. In addition, the following sentence will be added in the conclusion : " the Matlab code of the genetic algorithm is available on request from the authors".*

*The section 3 & 3.1 is replaced by this one :*

Computer-assisted variable selection is important for several reasons. Indeed, the selection of a subset of variables in a high dimension space can improve the performance of the model or its statistical properties, but also provides more robust models and reduce their complexity. In practice it is generally impossible to try all possible combinations of variables and select the best ones because of the computational cost. Many approaches exist for variables selection (Guyon and Elisseeff, 2003). Among then we choose to develop a model based on genetic algorithms to search for the best variables subset. Genetic algorithms (GAs) (Holland, 1975) are stochastic optimization algorithms based on the mechanics of natural selection and genetics described by Charles Darwin. In our study, a chromosome is defined as a subset of the 23 variables. A first generation composed of a population of 60 potential chromosomes is arbitrarily chosen. The performances of each chromosome (i.e. for each 60 corresponding subset of variables) is evaluated through a fitness function *f*. The fitness function is defined in such a way that the higher it is, the more the fitness function is able to represent the whole dataset (of dimension 23) with a minimal number of variables. Based on the performances of the 60 chromosomes we create a new generation of 60 chromosomes of potential solutions using classical evolutionary operators: selection, crossover and mutation. The performances of this new generation is then evaluated. This cycle is repeated until the stop criteria is asserted. The best chromosome of the last generation provides the optimal subset of variables.

**3.1 Methodology**

Let's define by $x^k$ the chromosome number k : $x^k = (x_1, x_2, ..., x_{23})$. $x^k$ is a binary vector in $\{0,1\}^{23}$ space such that each component has the following meaning:

$$\forall i \in \{1, ..., 23\}, \begin{cases} x_i = 1 & \text{The variable number } i \text{ in } Tab.2 \text{ column 1 is selected} \\ x_i = 0 & \text{The variable number } i \text{ in } Tab.2 \text{ column 1 is not selected} \end{cases} \quad (1)$$

The word "selected" in eq. 1 means that the corresponding variable will be used both in the training step described just below in "Step 2" and for performances evaluation. Otherwise if the corresponding variable is not selected it will be used only for performances evaluation.

As previously stated the fitness function allows to provide a measure of how well a minimal subset of variables can

represent the entire data space (in dimension 23). To do that the fitness function f is defined as follow :

$$f(x^k) = \frac{1}{nb(x^k)\, te(x^k)}$$

(2)

With

$x^k$ : chromosomes number $k$
$nb(x^k)$ : number of selected variables in chromosome $x^k$
$te(x^k)$ : topological error associated to chromosome $x^k$

The aim is to minimise the number of selected variables *nb* and the topological error *te*. Consequently we seek to maximize the fitness function. The estimation of the topological error made from a Self-Organizing Maps (SOM) is somewhat complicated and requires some explanation. Self-Organizing Maps introduced by Teuvo Kohonen (Kohonen, 1982, 2001) is a popular clustering and visualization algorithm. SOM is a neural network algorithm that is based on unsupervised learning based on competitive learning (Kohonen, 1982, 2001; Vesanto and Alhoniemi, 2000). It may be considered has a nonlinear generalization that has many advantages over the conventional features extraction methods such as Empirical Orthogonal Functions (EOF) or Principal Component analysis PCA (e.g., Liu et al., 2006). SOM applications are becoming increasingly useful in geosciences (e.g., Liu and Weisberg, 2011). As was written by Uriarte and Martín (2008) : "The SOM provides a non-linear, ordered, smooth mapping of high-dimensional input data manifolds onto the elements of a regular, low-dimensional array. The main characteristic of the projection provided by the algorithm is the preservation of neighbourhood relations; as far as possible, nearby data vectors in the input space are mapped onto neighbouring locations in the output space". This property allows to compute easily a topological error (see Uriarte and Martín, 2008, eq. 2). For each of the $x^k$ chromosomes a Self-Organizing Maps $M(x^k)$ is learned on the training data set. Only the selected variables are used during the learning process. Finally for each Maps $M(x^k)$ the topological error $te(x^k)$ is then computed according to eq. 2 in Uriarte and Martín (2008). Section 4 provides additional information concerning Self-Organising Maps.

The Genetic Algorithm is based on the following five steps (Fig.2.):

*Step 1- Initialization:* (initial population)
Generate randomly, a population {$x^k$, k=$1, ..., 60$} of 60 chromosomes of dimension 23.
*Step 2- Evaluation:* For each of the $x^k$ chromosomes a SOM $M(x^k)$ is learned. Only the selected variables are used for learning. Once training each Maps provides the topologic error $te(x^k)$ allowing to calculate their fitness score $f(x^k)$ by the mean of the fitness function.
*Step 3-* Select the best chromosome $x^{Best}$ among the 60 chromosomes according to its fitness score previously computed on the test dataset. If $x^{Best}$ remains the same during 50 generations then stop the procedure and select the relevant variables, i.e. the ones for which the corresponding components are equal to 1 in $x^{Best}$. Else go to step 4.
 *Step 4- Selection:* Create a new population of 60 chromosomes from the current population by random sampling with replacement of chromosomes based on their probabilities calculated according to the formula:

$$\Pr(x^k) = \frac{f(x^k)}{\sum_{i=1}^{60} f(x^i)}$$

(3)

*Step 5- Reproduction:* Mutation and Crossover possibilities in the new population.
Mutation: It consists of modifying (or not) some components of the chromosomes. The probability of mutation is in general very low and is commonly set to $p = 10^{-7}$. In our case the number of necessaries generations to reach our objective is lower than a few hundred. Consequently in our case, the probability of a mutation is highly unlikely.
Crossover: First, we randomly draw $\frac{60}{2} = 30$ couples of chromosomes from our population. Then, for each couple ($x^k$, $x^l$) (called parents) one crossover point, noted $I_c$, is randomly drawn in the range [1, 23] using a discrete uniform law. Two new chromosomes ($x^{k'}$, $x^{l'}$) are created in the following way:

$$\begin{cases} x^{k'} = (x_1^k, x_2^k, ..., x_{I_c}^k, x_{I_c+1}^l, x_{I_c+2}^l, ..., x_{23}^l) \\ x^{l'} = (x_1^l, x_2^l, ..., x_{I_c}^l, x_{I_c+1}^k, x_{I_c+2}^k, ..., x_{23}^k) \end{cases}$$

(4)

So from two parents we generate two children allowing having a new generation with the same number of chromosomes. Finally go to step 2.

**Figure 2: Diagram for the selection of variables based on a Genetic Algorithm associated with Kohonen Maps and a fitness function**

*In Section 4 : Some additional information are provided. The following sentences replace lines 11 to 27 p7*

A SOM is a topological map composed of neurons. In our case, a neuron is a vector of dimension 23 containing the 23 variables defined previously in Tab. 2. Each neuron has 6 neighboring neurons. SOM is an unsupervised neural network trained by a competitive learning strategy that performs two tasks: vector quantization and vector projection. Different from K-means, SOM uses the neighborhood interaction set to learn the topological structure hidden in the data. In addition to the best matching referent vector (neuron), its neighbors on the map are updated, resulting in regions where neurons in the same neighborhood are very similar. It can be considered as an algorithm that maps a high dimensional data space to a two dimensional space called a map. A map can be used at the same time both to reduce the amount data by clustering and for projecting the data in a non-linearly way to a regular grid (the map grid).

In this study we used the toolbox developed by "the SOM Toolbox Team" available at the following address: http://www.cis.hut.fi/somtoolbox/. A SOM with 8×8 = 64 neurons is considered here. This choice corresponds to a compromise. A smaller map would not be able to distinguish fine details, whereas a large map would not make sense given the number of observations available.

After training by the GA algorithm described the previous section the obtained map $M(x^{Best})$ can be used to affect to any event the best matching referent vector (neuron) according to the 5 selected variables associated to the chromosome $x^{Best}$. Hence the obtained $M(x^{Best})$ map can be considered as an optimal representation of the initial data set.

Figure 3 shows the distances matrix. For each neuron the color indicates the mean distance between a neuron and its neighbors. The value at the center of each neuron represents the number of rain events of the training data set captured by the corresponding neuron. All neurons capture rain events and a bit more than half of them capture between 3 and 5 rain events which is close to the value that would be obtained ($234 / 64 \cong 4$) if the rain events were uniformly distributed on the map.

**The following references were added :**

Guyon, I. and André Elisseeff; An Introduction to Variable and Feature Selection    (Kernel Machines Section)3(Mar):1157--1182, 2003.

E. Arsuaga Uriarte, and F. Díaz Martín, Topology Preservation in SOM, World Academy of Science, Engineering and Technology. International Journal of Computer, Electrical, Automation, Control and Information Engineering Vol:2, No:9, 2008

Kohonen, T. (2001). Self-Organizing Maps. Springer-Verlag, ISBN 3-540-67921-9, New York, Berlin, Heidelberg

Vesanto, J. & Alhoniemi, E. (2000). Clustering of the self-organizing map. IEEE Transactions on Neural Networks, Vol. 11, 586–600, ISSN 1045-9227

Liu, Y.; Weisberg, R. H. & Mooers, C. N. K. (2006). Performance evaluation of the self- organizing map for feature extraction, Journal of Geophysical Research, Vol. 111, C05018, doi:10.1029/2005JC003117, ISSN 0148-0227

Liu, Y.,and R.H. Weisberg (2011) A review of self-organizing map applications in meteorology and oceanography. In: Self-Organizing Maps-Applications and Novel Algorithm Design, 253-272.

Specific Comments:

In addition to the presentation concerns outlined above (and by the other reviewers), I do have some basic scientific questions to add to the discussion.

1. As noted by other reviewers, I am curious as to how much the choice of a 30-minute MIT interval used to define a rain event effects the results. Similarly, how much does the minimum detection threshold influence the results?
*Our choice was made based on the literature. Even if it cannot be called a sensitivity study,  we also run some test to see if the conclusion drawn in the lite nevertheless literature were coherent with our data set.*
*We have not really investigated this point and consequently we are not really in position to answer to this question. A greater MIT value will aggregate rain periods together and consequently will modify some parameters like the event duration De or the Rain event Depth Rd. Some others parameters can remain unchanged or not, this is the case for example for parameters Pci (which is sensitive to the variability of the rain rate). We expect that a higher MIT may aggregate events of a different type, whereas a shorter MIT tends to increase the number of events while they belong to the same group of rain cells.*

*The chosen threshold (0.1 mm/h) allows to detect very light rain while avoiding false detections due to dust or insects. This threshold can play an important role concerning the rain support but has very little influence on whole of the others paramaters used in this study*

2. Given that this method is new to some readers, could it briefly be described what happens with this method if it is

employed with non-transformed data? The normalization method used to make each of the variables pseudo-normal sounds quite practical, but I always wonder what biases this can introduce in fundamental non-linear processes like rainfall. What would happen if such a transformation is not utilized?

*We used 7 transformations. Among the seven (see Table 3) 6 transformations are nonlinear. There are simply used to contract/dilate the data space to guarantee a better spreading of the map among the data. Without any transformation the resulting map would be less representative of the data.*

*Yes there are quite practical, properly speaking, there is no statistical or probabilistic framework in this article (No estimator, no confidence interval or no whatsoever are involved.)*

*The nonlinear transformations are intended to avoid any biais.*

3. The instrument you utilized had a 1-minute integration time. There are other detectors with sub-second integration times, and one can always coarsen data. What influence does integration time have on the method proposed? Does one find the same basic results (for the entire process if the full analysis is done with a different integration time-scale? This is not merely of academic interest here, since scale-matching is of vital importance within the hydrological community where instruments with very coarse resolution (e.g. radar) are often "ground-truthed" with point-detector measurements that can have sub-millisecond resolutions.

*From a practical point of view we could simply aggregate the data if they are at a resolution finer than 1 minute. Otherwise :*

*Almost the 23 parameters are derived from rain rate parameter. Hence the vectors representing an events will be different from the ones at a 1-minute resolution. We expect to find the same basic results conditioned on the loss of information due to a coarser resolution. For resolution closed to 1 minute the expected results should be close.*

*It is one of the perspectives of this work. It would give access to more exhaustive (as well in space and in time) data sets. We are working on this aspect. It has to be noticed that the nature of the measurements is different. (For example, a tipping bucket rain gauge is measuring the time to fill a given volume. It has some important implications for stratiform rain events.)*

*Since the resolution is less good it will definitely impact the results. It will strongly impact some parameters such as the Rmax and the PC parameters. Since Rmax and $PC_1$ are two of the five parameters selected by the algorithm it will impact the results. This is nevertheless a study work in itself.*

4. As noted by other reviewers, the conclusions seem a bit of an over-reach for a study done using one type of instrument, with one type of MIT choice, one minimum detection threshhold, at one location, with one integration time. I believe that if this approach was applied by a number of investigators at different locations using data from a number of different instruments and a variety of parameter choices it is possible that a very powerful result could be attained. I would love to work on such a study, but – due to the quality of written English in this paper – I can't follow the method closely enough to do so at this time.

I would like to petition the authors to give this paper a substantial re-write and have native English speakers edit it. In the interest of readability, I would encourage figure captions to be more descriptive. Finally, I would love to have access to either code or other materials necessary to replicate the work (so, as other reviewers have pointed out, unspecified details like the choices made in te(x^k) are known to a practitioner).

If these changes could be implemented, I would promise the authors that not only will this paper be read, but some of us will use the results in their own work – myself included.

*We agree. A professional translator will edit the manuscript. Additional information is now provided in sections 3 and 4. References are added to help to replicate the work.*

---

## Author Comment (AC5) · 6 Feb 2017

The authors present an interesting study on data driven clustering of rain events based
on the analysis of characteristics related to rain rates or rain accumulation (macro
physical information) and to the raindrop size distribution (microphysical information).
The research results are interesting, relevant and timely.
Some questions, and possibly suggestions for future work, are however presented

hereafter:

1) the authors adopt a Minimum Inter-event Time (MIT) of 30 minutes. Did they perform
any sensitivity of the presented results versus lower (15 minutes) and higher (60 and
120 minutes) threshold?

*Indeed, the study of the sensitivity to MIT has to be done, but regarding the future work our priority is to extend the study to measurements carried out by rain gauge with a time resolution of 5 mn (more common and accessible)*
*We have not really investigated this point and consequently we are not really in position to answer to this question. A greater MIT value will aggregate rain periods together and consequently will modify some parameters like the event duration De or the Rain event Depth Rd. Some others parameters can remain unchanged or not, this is the case for example for parameters Pci (which is sensitive to the variability of the rain rate). We expect that a higher MIT may aggregate events of a different type, whereas a shorter MIT tends to increase the number of events while they belong to the same group of rain cells*

2) a total of 23 indicators for macro physical description of rain events were defined,
however the references provided in order to support their choices are rather limited.
Please improve.

**Line 161 section 2.1 added references:**

**Haile 2011 Brown 1985 Dris 1989**

**Added reference :**

Driscoll, E. D., Palhegyi, G. E., Strecker, E. W., & Shelley, P. E. (1989). Analysis of storm events characteristics for selected rainfall gauges throughout the United States. *US Environmental Protection Agency, Washington, DC*.

**Most parameters (mean, maximum, standard deviation,...) are common statistical descriptors for witch there is no particular references.**

3) also the details provided about the 23 indicators computation are rather poor. Please
improve

*Among the 23 indicators described some of them are very classical like the event duration or Rain event depth. Some are less classical they are described in the provided references. We voluntarily chose not to detail certain indicators in order to shorten the paper.*

4) by checking the position of the test site, namely "Site Instrumental de Recherche par Télédétection Atmosphérique" (SIRTA1) in Palaiseau (France), i noticed that it is not far from the Trappes sounding site. Then i was wondering if the authors plan to explore possible relationship between the identified characteristics related to rain rates or rain accumulation (macro physical information) and to the raindrop size distribution (microphysical information) versus the vertical thermodynamical structure observed for the identified events in the period 2008-2014. For example

Molini, L., Parodi, A., Rebora, N., Craig, G. C. (2011). Classifying severe rainfall
events over Italy by hydrometeorological and dynamical criteria. Quarterly Journal of
the Royal Meteorological Society, 137(654), 148-154.

For about 81 events, a time-scale for convective adjustment was computed, based on gridded hourly precipitation rates derived from rain-gauge data and ECMWF analysis (ERA-Interim) of convective available potential energy (CAPE). Values of the convective adjustment time-scale, $\tau$ c, shorter than 6 h indicate convection that is responding rapidly to to the synoptic environment (equilibrium), while slower time-scales indicate that other, presumably local, factors dominate.

It would be interesting to see if and how a local convective adjustment time-scale (computed using Trappes data) is related to the results of this study.

**added p10 l25:**

**The hypothesis that the two categories of precipitation events corresponding to different dynamical regimes can be identified solely on the basis of hydrometeorological variables confirms the study of Molini et al (2011). These authors have shown the agreement between the hydrometeorological classification (based on duration and extent of events from rain gauge network data) and dynamical classifications (the convective adjustment time-scale identified to distinguish between equilibrium and non-equilibrium convection from ECMWF analysis)**
**Added reference :**
**Molini, L., Parodi, A., Rebora, N., Craig, G. C. (2011). Classifying severe rainfall events over Italy by hydrometeorological and dynamical criteria. Quarterly Journal of the Royal Meteorological Society, 137(654), 148-154.**

5) in a recent paper

Bühl, J., Leinweber, R., Görsdorf, U., Radenz, M., Ansmann, A., Lehmann, V. (2015). Combined vertical-velocity observations with Doppler lidar, cloud radar and wind profiler. Atmospheric Measurement Techniques, 8(8), 3527-3536.
it is explored the potential of combined vertical-velocity observations with Doppler lidar, cloud radar and wind profiler.
In this respect, even if the sodar located at the Charles de Gaulle Airport is relatively far away (50 km), did the authors consider as possible to explore relationship (if any) between updraft and downdraft velocity versus the microphysical analysis performed in this study?
I would be very curious to test, in a real-word situation, the results we got years ago in these papers about the relationship between raindrop diameter and updraft velocities:
Parodi, A., Emanuel, K. (2009). A theory for buoyancy and velocity scales in deep moist convection. Journal of the Atmospheric Sciences, 66(11), 3449-3463.
Parodi, A., Foufoula Georgiou, E., Emanuel, K. (2011). Signature of microphysics on spatial rainfall statistics. Journal of Geophysical Research: Atmospheres, 116(D14).

**We are agree with the reviewer that the demonstration of the existence of a relationship between hydrometeorological variables and atmospherical processes can be exploited by carrying out the joint analyses of the two types of observations. We have thus explore the potential of combined measurements in Mercier et al. 2016 . We have built a 4-D-VAR data assimilation algorithm for retrieving vertical DSD profiles and vertical wind under the bright band from observations coming from a micro-rain radar (MRR) and from a co-located disdrometer, associated with a vertical advection model. This is also intended as a tool to better characterize rainfall microphysical processes. The coupling is done via a 4-D-VAR data assimilation algorithm. The dynamical model and the geometry of the problem are quite simple. They do not allow for the moment the complexity implied by all rain microphysical processes to be encompassed (evaporation, coalescence breakup and horizontal air motion are not taken into account). In the end, the model is limited to the fall of droplets under gravity, modulated by the effects of vertical winds et by the evaporation. Because of the limitations of the model, the retrieval algorithm is currently only suitable to study stratiform, light rain events. Some improvements are needed to provide an algorithm suitable for various weather situations (tropical rain and convective events, for instance). This is an ongoing work**

**Mercier F., Chazottes A., Barthès L., Mallet C. 4-D-VAR assimilation of disdrometer data and radar spectral reflectivities for raindrop size distribution andvertical wind retrievals Atmospheric Measurement Techniques, European Geosciences Union, 2016, 9 (7), pp.3145-3163. <10.5194/amt-9-3145-2016> - insu-01233557**